# Posterior Concentration of Bayesian Physics-Informed Neural Networks for Elliptic PDEs

Yuxuan Zhao [1]    Yulong Lu [1]

## Abstract

We study the posterior contraction rate of Bayesian Physics-Informed Neural Networks (PINNs) for solving a general class of elliptic partial differential equations (PDEs). We focus on learning of the elliptic equation with a non-homogeneous Dirichlet boundary condition from independent and noisy measurements collected both inside the domain and on the boundary. Assuming that the PDE admits a strong solution in a Hölder space and using with a suitably constructed prior on the neural network weights, we prove that the posterior distribution concentrates around the exact solution at a near-minimax rate. Furthermore, the chosen prior is *rate-adaptive*: the posterior contracts at an (almost) optimal rate without prior knowledge of the smoothness level of the exact solution. Our results provide statistical guarantees for uncertainty quantification of PDEs via Bayesian PINNs.

## 1. Introduction

Deep neural networks (DNNs) or multi-layer perceptrons (MLPs) offer various inherent advantages over traditional approaches of scientific computing and data analysis, such as finite element methods, wavelets and kernel methods, which are often hampered by the irregular and nonlinear data structures and the high input dimensions. In contrast, DNNs are capable of approximating a rich class of functions with aforementioned complexities and can also easily encodes additional complex physical structures, such as symmetry and other invariant structures. This enables DNNs to tackle outstanding challenges in science and engineering that are modeled by partial differential equations (PDEs). A burgeoning line of recent research has explored neural

networks as an ansatz for approximating PDE solutions (Yu et al., 2018; Raissi et al., 2019; Sirignano & Spiliopoulos, 2018; Zang et al., 2020), offering algorithmic innovations for solving PDEs with extremely promising numerical performances. In this paper, we focus on Physics-Informed Neural Networks (PINNs) (Raissi et al., 2019) due to their great flexibility and versatility. The core idea of PINNs is to enforce PDE constraints through minimizing a training loss defined by the residual of the governing equations. From a statistical perspective, PINNs produce point estimators of the PDE solutions. However, in practice, due to experimental and simulation constraints, the parameters of PDE models (e.g. internal and boundary data) are often subject to considerable uncertainties, because they can be prohibitively costly to obtain fully or to carry out exact discrete evaluations. These model uncertainties and discretization uncertainties, if assumed negligible, can lead to serious misspecification. In this setting, understanding how those uncertainties propagate to the neural-network solutions is of crucial importance to quantify the robustness of the deep learning-based PDE solvers.

We adopt a Bayesian perspective on the uncertainty quantification of the neural-network solutions of elliptic PDEs. Unlike a standard PINN—which produces an approximate solution by minimizing a PDE-residual loss and thus yields only a point estimate, failing to quantify uncertainty induced by noisy or limited data, a Bayesian PINN returns a full posterior distribution over solutions by combining the uncertain information from the likelihood (data) and the prior. Bayesian neural networks, originating in the seminal works of MacKay (MacKay, 1995) and Neal (Neal, 1995), have been extensively studied over the past three decades (Lampinen & Vehtari, 2001; Titterington, 2004; Graves, 2011). Recently, they have been coupled with PINNs for forward and inverse problems in differential equations (Yang et al., 2021); however, rigorous theoretical guarantees for Bayesian PINNs remain limited.

### 1.1. Summary of contribution

In this paper, we establish the posterior concentration rate of Bayesian PINNs for a general class of elliptic equations, i.e. identifying the rate at which the posterior distribution

[1]School of Mathematics, University of Minnesota, Twin Cities, 127 Vincent Hall 206 Church St. SE Minneapolis, MN 55455. Correspondence to: Yulong Lu <yulonglu@umn.edu>.

*Proceedings of the 43rd International Conference on Machine Learning*, Seoul, South Korea. PMLR 306, 2026. Copyright 2026 by the author(s).

on the solution shrinks towards the exact solution in the large sample limit. With a carefully chosen prior distribution on the neural network weights, we prove that when the exact solution is $\beta$-Hölder smooth, the posterior contracts at a near–minimax-optimal rate with respect to the norm induced by the PINN-loss. It is worth-noting that the resulting Bayesian PINN is rate-adaptive: it achieves the optimal contraction rate without prior knowledge of the unknown solution's smoothness.

## 1.2. Related work

**Convergence analysis of vanilla PINNs** The approximation power and statistical performance of DNNs for PDEs have been studied extensively in recent years within variational frameworks such as PINNs (Mishra & Molinaro, 2023; De Ryck & Mishra, 2022; Shin et al., 2023; Lu et al., 2021a) and the Deep Ritz Method (E & Yu, 2018; Jiao et al., 2024). Under Sobolev or Hölder smoothness assumptions on the PDE solution, these works derive quantitative, nonparametric estimation rates that generally suffer from the curse of dimensionality. More recently, a complementary line of research (Chen et al., 2021; 2023; Weinan & Wojtowytsch, 2022; Lu & Lu, 2022; Lu et al., 2021b) established dimension-independent rates for certain high-dimensional PDEs by imposing Barron-space regularity. Among the aforementioned works, (Lu et al., 2021a) is closest to our setting: it proves minimax-optimal estimation rates for both PINNs and the Deep Ritz method for a class of elliptic PDEs. However, Our contribution differs in several key respects. First, we establish posterior contraction rates for Bayesian PINNs, whereas (Lu et al., 2021a) analyzes point estimators (effectively maximum-likelihood/ERM-type procedures under the chosen variational losses). Second, with an appropriate prior, our procedure is adaptive to the (unknown) smoothness of the exact solution, while the rates in (Lu et al., 2021a) are non-adaptive. Finally, our data model includes noisy measurements of both the interior source term and a nonhomogeneous Dirichlet boundary condition, and the resulting PINN loss penalizes both interior and boundary residuals. In contrast, (Lu et al., 2021a) assumes homogeneous Dirichlet boundary condition, ignores the boundary residue in their PINN loss, and enforces the neural networks vanish entirely on the boundary, which is challenging to satisfy in practice.

**Bayesian neural networks and posterior contraction** Bayesian neural networks (MacKay, 1995; Neal, 1995) provide a principled framework for uncertainty quantification of neural-network predictors. This approach combines the expressive power of DNNs and probabilistic reasoning of Bayesian inference leading to strong empirical generalization capability and superior uncertainty estimates (Izmailov et al., 2021; Wilson & Izmailov, 2020). Despite the re-markable empirical successes of Bayesian neural networks, their theoretical understandings are relative scattered. Building on the DNN approximation theory of (Schmidt-Hieber, 2020), the pioneering work of (Polson & Ročková, 2018), proved that the posterior distribution of Bayesian neural networks with a class of spike-and-slab priors concentrate at the minimax rate for nonparametric regression of $\beta$-Hölder smooth functions. (Kong et al., 2023) obtained a similar contraction result for masked Bayesian neural networks, which allows more efficient computation in practice. The work by (Kong & Kim, 2025) extended the posterior contraction result of (Polson & Ročková, 2018) using non-sparse Gaussian priors. The work by (Lee & Lee, 2022) proved the posterior contraction rate for Besov functions using a class of shrinkage priors. These results were further extended for anisotropic and composite Besov functions using heavy-tailed priors (Egels & Castillo, 2025) or spike-and-slab priors and shrinkage priors (Lee et al., 2025). We also note that variational Bayes (Blei et al., 2017) is widely used as a tractable alternative to exact posterior inference (e.g., mean-field approximations), and recent results show that the resulting variational posteriors can achieve the same near-optimal rates (Chérief-Abdellatif, 2020; Bai et al., 2020; Ohn & Lin, 2024).

While posterior contraction for Bayesian neural networks is now well developed for classical nonparametric regression, little is understood for physics-informed learning of PDEs. The goal of the present work is to fill this gap by establishing the posterior contraction rate for Bayesian PINNs.

**Bayesian PINNs** In the empirical study of (Yang et al., 2021), a Bayesian perspective of PINN was proposed to quantify the uncertainty of the neural network solution obtained by PINN, with application spanning real-world dynamical systems (Linka et al., 2022), turbulence modeling (Shukla et al., 2026), and subsurface seismic tomography (Gou et al., 2023). In particular, the posterior consistency was observed numerically (see Figure 4 of (Yang et al., 2021)), but no theoretical analysis was provided. The recent paper (Sun et al., 2024) is most closely related to ours, which established posterior contraction rates for Bayesian PINNs when estimating the PDE solution along with a finite number of unknown finite-dimensional parameters, using independent noisy observations of the solution itself. Our setting differs from (Sun et al., 2024) in a crucial aspect: we observe only the source term and the nonhomogeneous boundary data, rather than the solution. This indirect observation model leads to a statistically harder problem; consequently, although our rates are minimax optimal under this data model, they are necessarily slower than those in (Sun et al., 2024).

## 2. Preliminaries

### 2.1. Notation

Given a real-valued function $f : \Omega \to \mathbb{R}$ on a bounded domain $\Omega \subset \mathbb{R}^d$ and $1 \leq p < \infty$, we denote $\|f\|_{L^p(\Omega)} = (\int_\Omega |f(x)|^p dx)^{1/p}$. Similarly, consider $g : \partial\Omega \to \mathbb{R}$, and denote $\|g\|_{L^p(\partial\Omega)} = (\int_{\partial\Omega} |g(x)|^p d\sigma(x))^{1/p}$, where $\sigma$ is the surface measure on the boundary $\partial\Omega$. Denote $|\Omega|$ as the volume of $\Omega$ and $|\partial\Omega|$ as the surface measure of $\partial\Omega$. For a multi-index $\boldsymbol{\alpha} = (\alpha_1, \ldots, \alpha_d) \in \mathbb{N}_0^d$, we denote $|\boldsymbol{\alpha}| := \sum_{i=1}^d \alpha_i$ and denote the partial derivative as $\partial^{\boldsymbol{\alpha}} := \partial^{\alpha_1} \ldots \partial^{\alpha_d}$. For $N \in \mathbb{N}$, denote $[N] := \{1, \ldots, N\}$. For a vector $b \in \mathbb{R}^d$, denote the $l_p$-norm as $\|b\|_p$ for $1 \leq p \leq \infty$, and let $\|b\|_0$ be the number of nonzero entries of $b$. Given a matrix $A \in \mathbb{R}^{d \times d}$, let $\|A\|_\infty := \max_{i,j \in [d]} |A_{ij}|$ and $\|A\|_0$ be the number of nonzero entries of $A$, and define $\nabla A \in \mathbb{R}^d$ by $(\nabla A)_i := \sum_{j=1}^d \partial_j A_{ji}$ if $A$ is differentiable. We use $\mathrm{Ber}(p)$ to represent a Bernoulli distribution with a success probability $p$ and $\mathrm{U}(I)$ for a uniform distribution on an interval $I$ respectively. Denote $\mathcal{N}(\mu, \sigma^2)$ as the Gaussian distribution with mean $\mu$ and variance $\sigma^2$. Given a probability distribution $P$ and a $P$-measurable function $f$, we note $P(f) := \int f dP$. We use the notation $a_n \lesssim b_n$ (resp. $a_n \gtrsim b_n$), or equivalently $a_n = O(b_n)$ to indicate that $a_n \leq C b_n$ (resp. $a_n \geq C b_n$) for some universal constant $C > 0$ independent of $n$. We use the notation $a_n = o(b_n)$ to denote that $\frac{a_n}{b_n} \to 0$ as $n \to \infty$. For a sequence of random variables $\{X_n\}_{n \in \mathbb{N}}$, we write $X_n = o_p(1)$ to mean that $X_n \to 0$ in probability as $n \to \infty$.

**Deep neural network spaces**   We denote the ReLU activation function as $\sigma_0(x) := \max\{x, 0\}$, and the ReLU powers as $\sigma_k(x) := (\max\{x, 0\})^k$. The parameter space of $L$-layer $W$-width neural networks with $B$-bounded and $S$-sparse weights is defined as

$$
\begin{aligned}
&\Theta(L, W, S, B) := \\
&\{\theta = (W^{(1)}, b^{(1)}, \ldots, W^{(L)}, b^{(L)}) : \\
&W^{(1)} \in \mathbb{R}^{d \times W}, b^{(1)} \in \mathbb{R}^W, \\
&W^{(l)} \in \mathbb{R}^{W \times W}, b^{(l)} \in \mathbb{R}^W, l = 2, \ldots, L-1, \\
&W^{(L)} \in \mathbb{R}^{W \times 1}, b^{(L)} \in \mathbb{R}, \\
&\max_{l \in [L]} \{\|W^{(l)}\|_\infty, \|b^{(l)}\|_\infty\} \leq B, \\
&\sum_{l=1}^L \|W^{(l)}\|_0 + \|b^{(l)}\|_0 \leq S\}.
\end{aligned}
$$

Let $f_\theta(\cdot) := (W^{(L)} \sigma_k(\cdot) + b^{(L)}) \circ \cdots \circ (W^{(2)} \sigma_k(\cdot) + b^{(2)}) \circ (W^{(1)}(\cdot) + b^{(1)})$ be the output function of a neural network with parameter $\theta$ and $\sigma_k$ activation function. In this work, we only consider DNNs with $\sigma_3$ activation functions. Then we can define the DNN-space consisting of functions $f_\theta$

with $\sigma_3$ being the activation functions, i.e.

$$
\mathcal{F}(L, W, S, B) := \{\mathrm{clip} \circ f_\theta : \theta \in \Theta(L, W, S, B)\},
$$

where clip is a $C^2$-cutoff function implemented by a single-layer $\sigma_3$-network, and the detailed construction can be found in Appendix D. With the clip function, our function class becomes uniformly bounded, which plays a critical role in bounding the statistical complexity of the network class.

Let $T$ to be the total number of parameters of a function $f_\theta$ with $\theta \in \Theta(L, W, S, B)$, i.e.

$$
T := dW + W + (L-1)(W^2 + W) + W + 1,
$$

and denote $\gamma \in \{0, 1\}^T$ as the architecture (the location of those nonzero parameters), i.e. $\gamma_i = 1$ if and only if the $i$-th parameter is nonzero.

**Sobolev functions**   Let $1 \leq p < \infty$ and $k \in \mathbb{N}$. The Sobolev spaces $W^{k,p}(\Omega)$ consists of all functions $u : \Omega \to \mathbb{R}$ such that for each multi-index $\boldsymbol{\alpha}$ with $|\boldsymbol{\alpha}| \leq k$, the weak derivatives $D^{\boldsymbol{\alpha}} u$ exist and belong to $L^p(\Omega)$, i.e.

$$
W^{k,p}(\Omega) = \{u : \Omega \to \mathbb{R}; \|u\|_{W^{k,p}(\Omega)} < \infty\},
$$

where the Sobolev norm $\| \cdot \|_{W^{k,p}(\Omega)}$ is defined by

$$
\|u\|_{W^{k,p}(\Omega)} := \sum_{|\boldsymbol{\alpha}| \leq k} \|D^{\boldsymbol{\alpha}} u\|_{L^p(\Omega)}.
$$

Also, we define the Sobolev space $W^{k,\infty}(\Omega)$ as

$$
W^{k,\infty}(\Omega) = \{u : \Omega \to \mathbb{R}; \|u\|_{W^{k,\infty}(\Omega)} < \infty\},
$$

where the Sobolev norm $W^{k,\infty}(\Omega)$ is defined via essential supremum, i.e.

$$
\|u\|_{W^{k,\infty}(\Omega)} := \sum_{\boldsymbol{\alpha} \leq k} \operatorname{ess\,sup}_\Omega |D^{\boldsymbol{\alpha}} u|.
$$

**Hölder functions**   Let $\beta > 0$, the Hölder functions over some domain $\Omega$ are defined as

$$
C^\beta(\Omega) := \left\{ f : \Omega \to \mathbb{R}; \|f\|_{C^\beta(\Omega)} < \infty \right\},
$$

where $\|f\|_{C^\beta(\Omega)}$ denotes the Hölder norm defined by

$$
\begin{aligned}
\|f\|_{C^\beta(\Omega)} := &\sum_{\boldsymbol{\alpha} : |\boldsymbol{\alpha}| \leq q} \sup_{x \in \Omega} |\partial^{\boldsymbol{\alpha}} f(x)| \\
&+ \sum_{\boldsymbol{\alpha} : |\boldsymbol{\alpha}| = q} \sup_{x,y \in \Omega, x \neq y} \frac{|\partial^{\boldsymbol{\alpha}} f(x) - \partial^{\boldsymbol{\alpha}} f(y)|}{|x - y|^{\beta - q}},
\end{aligned}
$$

where $q$ is the greatest integer less than $\beta$.

### 2.2. Elliptic PDEs

We consider the following elliptic equation with Dirichlet boundary condition:

$$\begin{aligned}
-\mathrm{div}(A\nabla u) + Vu &= f \quad \text{in } \Omega, \\
u &= g \quad \text{on } \partial\Omega,
\end{aligned} \tag{1}$$

where for simplicity we assume that the domain $\Omega \subset \mathbb{R}^d$ is a bounded open subset with smooth boundary $\partial\Omega$. We assume that the coefficients $A$ and $V$ are sufficiently smooth, i.e. $A, V \in C^\infty(\bar{\Omega})$, and there exist some strictly positive constants $r_{\min}, C_A, V_{\min}, V_{\max}$ such that $r_{\min}I \preceq A$, $\sup_{x\in\Omega} \|A(x)\|_\infty \vee \|\nabla A(x)\|_\infty \leq C_A$, and $0 < V_{\min} \leq V \leq V_{\max}$ over $\Omega$. Moreover, we assume that $f \in C^{\beta-2}(\bar{\Omega})$ and $g \in C^\beta(\partial\Omega)$ with $\beta > 2$. The standard elliptic PDE theory (Grisvard, 2011) guarantees a unique strong solution $u^* \in C^\beta(\bar{\Omega})$ to (1).

### 2.3. Physics-Informed Neural Network

We focus on using Physics-Informed Neural Networks to solve (1). Specifically, for a constant $\lambda > 0$ and $u \in H^2(\Omega)$, we define the (population) PINN-loss by

$$\mathcal{E}(u) := \| -\mathrm{div}(A\nabla u) + Vu - f \|_{L^2(\Omega)}^2 + \lambda \|u - g\|_{L^2(\partial\Omega)}^2. \tag{2}$$

**Empirical PINN-loss**   Let $\{X_i\}_{i=1}^n$ be an i.i.d. sequence of random variables distributed according to the uniform distribution over $\Omega$, and let $f_i = f(X_i) + \epsilon_i$ be noisy observations of the right hand side of the PDE problem (1) with $\epsilon_i \overset{i.i.d.}{\sim} \mathcal{N}(0,1)$ being independent from $X_i$. Similarly, let $\{Y_j\}_{j=1}^n$ to be independently drawn of the uniform distribution over $\partial\Omega$, and set $g_j = g(Y_j) + \eta_j$ with $\eta_j \overset{i.i.d.}{\sim} \mathcal{N}(0,1)$ being independent from $Y_j$. Denote $\mathcal{D}_1^{(n)} := \{X_i, f_i\}_{i=1}^n$, $\mathcal{D}_2^{(n)} := \{Y_j, g_j\}_{j=1}^n$, and $\mathcal{D}^{(n)} := \mathcal{D}_1^{(n)} \cup \mathcal{D}_2^{(n)}$. The empirical counterpart of (2) is defined as

$$\begin{aligned}
&\mathcal{E}_n(u; \mathcal{D}^{(n)}) \\
&:= \frac{|\Omega|}{n} \sum_{i=1}^n | -\mathrm{div}(A\nabla u)(X_i) + V(X_i)u(X_i) - f_i|^2 \\
&\quad + \lambda\frac{|\partial\Omega|}{n} \sum_{i=1}^n |u(Y_j) - g_j|^2.
\end{aligned} \tag{3}$$

Moreover, we define the empirical distances

$$\begin{aligned}
&\| -\mathrm{div}(A\nabla(u-u^*)) + V(u - u^*) \|_{n,1}^2 \\
&:= \frac{|\Omega|}{n} \sum_{i=1}^n (-\mathrm{div}(A\nabla(u-u^*))(X_i) + V(X_i)(u-u^*)(X_i))^2 \\
&= \frac{|\Omega|}{n} \sum_{i=1}^n (-\mathrm{div}(A\nabla u)(X_i) + V(X_i)u(X_i) - f(X_i))^2,
\end{aligned} \tag{4}$$

and

$$\begin{aligned}
\|u - u^*\|_{n,2}^2 &:= \frac{|\partial\Omega|}{n} \sum_{j=1}^n (u(Y_j) - u^*(Y_j))^2 \\
&= \frac{|\partial\Omega|}{n} \sum_{j=1}^n (u(Y_j) - g(Y_j))^2.
\end{aligned} \tag{5}$$

### 2.4. Priors

We use a spike-and-slab prior $\Pi_\theta$ on the parameter space $\Theta := \Theta(L) = \cup_{W\geq 1} \cup_{S\geq 1} \cup_{B>0} \Theta(L, W, S, B)$. Specifically, we take $L = O(1)$ to be some fixed constant. For the width $W$, we assign

$$\Pi_\theta(W = w) = \frac{\lambda_W^w}{(e^{\lambda_W} - 1)w!}, \quad w = 1, 2, \ldots,$$

where $\lambda_W$ is some positve constant. Given $W = w$, let $T = O(W^2) = O(w^2)$ be the total number of parameters, and treat the architecture $\gamma$ as unknown with a Bernoulli distribution

$$\gamma_i | W = w \sim \mathrm{Ber}((1 + T^{\lambda_S})^{-1}), \quad i \in [T],$$

for some positive constant $\lambda_S$. In other word, the sparsity level $S$ is associated with a binomial prior

$$\begin{aligned}
&\Pi_\theta(S = s | W = w) \\
&= \binom{T}{s} \left(\frac{1}{1 + T^{\lambda_S}}\right)^s \left(1 - \frac{1}{1 + T^{\lambda_S}}\right)^{T-s} \\
&= \binom{T}{s} \frac{T^{\lambda_S(T-s)}}{(1 + T^{\lambda_S})^T}.
\end{aligned}$$

We further endow the size of the parameters with an exponential distribution

$$B \sim \mathrm{Exp}(\lambda_B),$$

for some $\lambda_B > 0$, and associate the prior on each entry of parameters as

$$\theta_i | (B = b, \gamma_i = 1) \sim \mathrm{U}([-b, b]), \quad \theta_i | \gamma_i = 0 \sim \delta_0.$$

The constants $\lambda_W, \lambda_S, \lambda_B$ will be determined later on.

Our choice of the spike-and-slab prior is inspired by (Polson & Ročková, 2018) but differs from theirs in several important aspects. First, we model sparsity by assigning each neuron an activation probability, which in turn induces a distribution over the overall sparsity level; by contrast, (Polson & Ročková, 2018) posits an exponentially distributed sparsity level directly. Second, we place an exponential prior on the amplitude $B$ of weights, thereby allowing unbounded weights, while (Polson & Ročková, 2018) restricts $B \leq 1$. The unboundedness of $B$ enables us to take advantage of

the approximation capacity of our DNN-class for the target solution (see Proposition 4.2), which will be essential to establish the key ingredients for the posterior contraction rate (see Lemma 4.1).

Recall the DNN-class $\mathcal{F} := \{\text{clip} \circ f_\theta : \bar{\Omega} \to \mathbb{R}; \theta \in \Theta\}$. Note that $\mathcal{F} \subset H^2(\bar{\Omega})$ since each function in $\mathcal{F}$ has $\sigma_3$ as activation functions. Furthermore, we equip $\mathcal{F}$ with trace sigma-algebra, denoted as $\Sigma_\mathcal{F}$, induced from the Borel sigma-algebra in $H^2(\bar{\Omega})$. Then we can define a prior $\Pi$ on $\mathcal{F}$ as

$$\Pi(E) := \Pi_\theta(\{\theta \in \Theta : \text{clip} \circ f_\theta \in E\}), \quad E \in \Sigma_\mathcal{F} \quad (6)$$

Note that for $r > 0$, the set $\{u \in \mathcal{F} : \| -\text{div}(A\nabla u) + Vu - f\|_{L^2(\Omega)} \leq r\} \in \Sigma_\mathcal{F}$ is measurable since the function $u \mapsto \| -\text{div}(A\nabla u) + Vu - f\|_{L^2(\Omega)}$ is continuous. Similarly, $\{u \in \mathcal{F} : \|u - g\|_{L^2(\partial\Omega)} \leq r\} \in \Sigma_\mathcal{F}$ is also measurable.

## 2.5. Posterior contraction for deep learning

Given the training data $\mathcal{D}^{(n)} = \mathcal{D}_1^{(n)} \cup \mathcal{D}_2^{(n)} = \{X_i, f_i\}_{i=1}^n \cup \{Y_j, g_j\}_{j=1}^n$, denote $p_u^{(n)}$ as the likelihood density of $\mathcal{D}_1^{(n)}$ under $u$, and $P_u^{(n)}$ as its corresponding distribution. Similarly, denote $q_u^{(n)}$ as the likelihood density of $\mathcal{D}_2^{(n)}$ under $u$, and $Q_u^{(n)}$ as its distribution. Then by Bayes' theorem, the posterior mass is given by

$$\Pi\left(E|\mathcal{D}^{(n)}\right) = \frac{\int_E p_u^{(n)} q_u^{(n)} d\Pi(u)}{\int_\mathcal{F} p_u^{(n)} q_u^{(n)} d\Pi(u)}, \quad E \in \Sigma_F.$$

We aim to establish the property of posterior contraction, which measures how fast the posterior mass concentrates around $u^*$ as $n \to \infty$. Following the standard literature on the general theory of posterior contraction rate (Ghosal & Van der Vaart, 2017, Chapter 8) and posterior contraction for deep learning (Polson & Ročková, 2018, Section 5), we define an $(M\epsilon)$-ball centered around $u^*$ by

$$A_{\epsilon,M} = \{u \in \mathcal{F} : \mathcal{E}(u) \leq M^2\epsilon^2\}.$$

Our goal is to show that for any $M_n \to \infty$

$$\Pi(A_{\epsilon_n, M_n}^c|\mathcal{D}^{(n)}) \to 0, \quad n \to \infty$$

in $P_{u^*}^{(n)} Q_{u^*}^{(n)}$-probability, or equivalently,

$$P_{u^*}^{(n)} Q_{u^*}^{(n)} \left[\Pi(A_{\epsilon_n, M_n}^c|\mathcal{D}^{(n)})\right] \to 0, \quad n \to \infty.$$

# 3. Main Results

## 3.1. Posterior contraction with rate adaption

We state our first main theorem that establishes the posterior distribution concentrates around the ground truth $u^*$ at a rate $\epsilon_n := n^{-\frac{\beta-2}{d+2(\beta-2)}}(\log n)^{1/2}$ without knowing the smoothness level of $u^*$.

**Theorem 3.1.** *Assume that $u^* \in C^\beta(\bar{\Omega})$ is the solution to (1) with $f \in C^{\beta-2}(\bar{\Omega})$ and $g \in C^\beta(\partial\Omega)$. Assume that the smoothness level $\beta$ satisfies that $2 < \beta \leq \beta^*$ for some $\beta^* > 2$, and $\|u^*\|_{C^\beta(\bar{\Omega})} \leq K$ for some $K > 0$. Then there exists a prior $\Pi$ of the form (6) on a DNN-space $\mathcal{F}$ such that*

$$\Pi(u \in \mathcal{F} : \mathcal{E}(u) > M_n^2\epsilon_n^2|\mathcal{D}^{(n)}) \to 0, \quad (7)$$

*in $P_{u^*}^{(n)} Q_{u^*}^{(n)}$-probability as $n \to \infty$ for any $M_n \to \infty$.*

**Corollary 3.2.** *Assume the same setting as in Theorem 3.1. Moreover, let $\Pi$ and the DNN-space $\mathcal{F}$ to be the same as in Theorem 3.1. Define the posterior mean as $\bar{u}_n := \mathbb{E}_{u \in \Pi}[u|\mathcal{D}^{(n)}]$. Then with high probability, $\mathcal{E}(\bar{u}_n) \lesssim \epsilon_n^2$ as $n \to \infty$.*

*Remark* 3.3. Thanks to the stability estimate of the PINN-loss shown in (Zeinhofer, 2022, Theorem 8), the contraction in PINN-loss implies the contraction in $H^{1/2}(\Omega)$-norm, i.e.

$$\Pi(u \in \mathcal{F} : \|u - u^*\|_{H^{1/2}(\Omega)} > M_n\epsilon_n|\mathcal{D}^{(n)}) \to 0,$$

in $P_{u^*}^{(n)} Q_{u^*}^{(n)}$-probability as $n \to \infty$ for any $M_n \to \infty$, and consequently we have the convergence of the posterior mean under $H^{1/2}$-norm $\|\bar{u}_n\|_{H^{1/2}(\Omega)} \lesssim \epsilon_n$ as $n \to \infty$. Unfortunately, the $L^2$-penalty on the boundary is too weak to ensure that the convergence in PINN-loss implies convergence in $H^s(\Omega)$ for any $s > 1/2$. Our result should be contrasted with the minimax rate for PINNs under $H^2$-norm in (Lu et al., 2021a) where they consider the homogeneous boundary condition and assume DNNs vanish on the boundary. In their homogeneous setting, $H^2$-norm is equivalent to the PINN-loss. In contrast, to obtain a convergence rate in $H^s(\Omega)$-norm with $s \geq 1$ in our non-homogeneous setting, it is necessary to use a stronger Sobolev norm $H^{s-1/2}$ on the boundary instead of the $L^2$-norm in the PINN-loss. However, approximating $H^{s-1/2}(\partial\Omega)$-norms on the boundary by discrete norms using only function values at the data sites is more involved. It is an interesting direction to develop feasible approaches for approximating the Sobolev norms boundary and study the resulting posterior contraction rate in possibly a stronger norm.

*Remark* 3.4. For convenience, we make the assumption in section 2.2 that $A, V \in C^\infty(\bar{\Omega})$ to ensure the regularity of the PDE solution. This assumption, however, can be relaxed. In particular, it suffices to assume that $A \in C^{\beta^*-1}(\bar{\Omega})$ and $V \in C^{\beta^*-2}(\bar{\Omega})$, under which there exists a unique strong solution $u^* \in C^\beta(\bar{\Omega})$ to (1) with $f \in C^{\beta-2}(\bar{\Omega})$ and $g \in C^\beta(\partial\Omega)$; see (Grisvard, 2011).

## 3.2. Minimax lower bound

We also study the minimax lower bound of PINN-loss given unequal sizes of noisy measurements taken both within the domain and from the boundary.

**Theorem 3.5.** *Assume $u^* \in C^\beta(\bar{\Omega})$ is a solution to (1) with $\beta > 2$, and $\|u^*\|_{C^\beta(\bar{\Omega})} \leq K$. For $n_1 \in \mathbb{N}$, let $\{X_i\}_{i=1}^{n_1}$ be*

an i.i.d. sequence of random variables distributed according to the uniform distribution over $\Omega$, and $f_i = f(X_i) + \epsilon_i$ with $\epsilon_i \overset{\text{i.i.d.}}{\sim} \mathcal{N}(0,1)$ being independent from $X_i$. For $n_2 \in \mathbb{N}$, let $\{Y_j\}_{j=1}^{n_2}$ be an i.i.d. sequence of uniformly distributed random variables over $\partial\Omega$, $Y_j = g(Y_j) + \eta_j$ with $\eta_j \overset{\text{i.i.d.}}{\sim} \mathcal{N}(0,1)$ being independent from $Y_j$. Denote $\mathcal{D} = \{X_i, f_i\}_{i=1}^{n_1} \cup \{Y_j, g_j\}_{j=1}^{n_2}$. Then we have

$$\inf_{\psi} \sup_{u^*} \mathbb{E}\mathcal{E}(\psi(\mathcal{D})) \gtrsim n_1^{\frac{2(\beta-2)}{d+2(\beta-2)}} + n_2^{\frac{2\beta}{d-1+2\beta}},$$

where the supremum is taken over all $u^* \in C^\beta(\bar{\Omega})$ which is a solution to (1) with $f \in C^{\beta-2}(\bar{\Omega})$ $g \in C^\beta(\partial\Omega)$ such that $\|u^*\|_{C^\beta(\bar{\Omega})} \leq K$, and the infimum is taken over all estimators $\psi : (\mathbb{R}^d)^{\otimes n_1} \times \mathbb{R}^{\otimes n_1} \times (\mathbb{R}^d)^{\otimes n_2} \times \mathbb{R}^{\otimes n_2} \to C^\beta(\bar{\Omega})$.

*Remark* 3.6. In view of Theorem 3.1 and minimax lower bound of Theorem 3.5 with $n_1 = n_2$, Bayesian PINN with an equal number of boundary and interior measurements attains a posterior contraction rate that is near–minimax optimal. Under our assumption that the true solution is Hölder continuous, 3.5 shows that any estimators, including PINNs, may suffer from curse of dimensionality, which is consistent with the literature (Lu et al., 2021a; Jiao et al., 2021). On the other hand, if the solution is assumed to have better regularity, e.g., if the solution is Barron assumed in (Lu et al., 2021b), then the generalization error of deep Ritz method does not incur a curse of dimensionality; their analysis can be adapted to PINNs under the same Barron assumption.

## 4. Proof Sketch

In this section, we outline the proof idea of the results in Section 3 and defer the complete proofs to the appendix.

### 4.1. Proof sketch of Theorem 3.1

The proof of Theorem 3.1 can be outlined as follows. Utilizing the theoretical framework developed by (Ghosal & Van Der Vaart, 2007), we first derive a posterior contraction rate with respect to the empirical PINN-loss $\mathcal{E}_n(\cdot; \mathcal{D}^{(n)})$; see Lemma 4.1. Then we transfer this contraction to the population PINN-loss by controlling the generalization gap (see Proposition 4.3).

**Posterior contraction with respect to the empirical distances** We first adapt the posterior contraction framework of (Ghosal & Van Der Vaart, 2007) to our setting and establish contraction of the posterior in the empirical PINN-loss for the PDE problem.

**Lemma 4.1.** *Assume $u^*$ satisfies the same assumption as in Theorem 3.1. Moreover, assume that there exists a subset*

$\mathcal{F}_n \subset \mathcal{F}$ and a constant $C > 0$ such that

$$\log \mathcal{N}(\frac{\epsilon_n}{4}, \mathcal{F}_n, \| -\text{div}(A\nabla \cdot) + V \cdot \|_{n,1}) \leq Cn\epsilon_n^2,$$
$$\log \mathcal{N}(\frac{\epsilon_n}{4}, \mathcal{F}_n, \| \cdot \|_{n,2}) \leq Cn\epsilon_n^2, \tag{8}$$

*and there exist a prior probability measure $\Pi$ on $\mathcal{F}$, and some constant $\xi > 0$, such that*

$$\Pi(u \in \mathcal{F} : \mathcal{E}_n(u; \mathcal{D}^{(n)}) \leq \epsilon_n^2) \geq \exp(-\xi n\epsilon_n^2), \tag{9}$$

*and*

$$\Pi(\mathcal{F}\backslash\mathcal{F}_n) = o(\exp(-(\xi+2)n\epsilon_n^2)). \tag{10}$$

*Then*

$$\Pi(u \in \mathcal{F} : \mathcal{E}_n(u; \mathcal{D}^{(n)}) > M_n^2\epsilon_n^2 | \mathcal{D}^{(n)}) \to 0,$$

*in $P_{u^*}^{(n)} Q_{u^*}^{(n)}$-probability as $n \to \infty$ for any $M_n \to \infty$.*

The entropy condition (8) controls the size $\mathcal{F}_n$ of the sieve while (10) guarantees that it should also receive the majority of the prior mass. The prior condition (9) ensures a non-vanishing mass in a shrinking neighborhood around the ground truth. The detailed proof of Lemma 4.1 can be found in Appendix A.1.

**Construction of the sieve** The construction of the sieve $\mathcal{F}_n$ relies on the following approximation result.

**Proposition 4.2.** *Let $2 < \beta$ and $l \in \mathbb{N}$. Then for any $f \in C^\beta(\bar{\Omega})$, there exists a $\sigma_3$-DNN $\phi$ with depth $L = O(\log \beta + \log d)$, width $W = O(d\beta \cdot l^d)$, sparsity $S = O(d\beta \cdot l^d)$, and size of parameters $B = O(l^d)$, such that*

$$\|f - \phi\|_{C^2(\bar{\Omega})} \lesssim l^{-(\beta-2)} \|u\|_{C^\beta(\bar{\Omega})}.$$

Proposition 4.2 is a restatement of (Lu et al., 2021a, Theorem 4.5). However, upon a careful inspection of the proof of (Lu et al., 2021a, Theorem 4.5), we notice that their proof is only presented for the smoothness level $\beta \leq 4$. We will provide a proof of the result for any $\beta > 2$. Moreover, their approximation error bound is quantified in terms of $W^{\beta,2}$-norm, while we control the error in a $C^\beta$-norm, which turns out to be crucial in proving posterior contraction under the empirical loss $\mathcal{E}_n(\cdot; \mathcal{D}^{(n)})$. Thanks to the standard interpolation theory, we do not require the smoothness level $\beta$ be integer and we characterize explicitly the dependence of neural network parameters on $\beta$ and $d$, which will also be critical in our design of the sieve neural network $\mathcal{F}_n$ and the overall set of networks $\mathcal{F}$. The proof of Proposition 4.2 can be found in Appendix A.2.

To construct the sieve $\mathcal{F}_n$, the idea is to ensure the existence of a function in $\mathcal{F}_n$ that approximates the ground truth within error $\epsilon_n$, which is a key ingredient in verifying the

condition (9), while controlling the size of $\mathcal{F}_n$ by enforcing (8). We define the sieve $\mathcal{F}_n$ as follows:

$$\mathcal{F}_n = \cup_{W=1}^{W_n} \cup_{S=1}^{S_n} \cup_{B \leq B_n} \mathcal{F}(L, W, S, B). \quad (11)$$

Specifically, we take $L = O(\log d + \log \beta^*)$ independent of $n$ with $\beta^*$ representing the maximum of smoothness level $\beta$ as in the assumption of Theorem 3.1. It is worth noting that $L$ depends only on $\Omega$ and $\beta^*$, rather on the approximation accuracy or the function to be approximated. Moreover, let $W_n = C_W n^{\frac{d}{d+2(\beta-2)}}$, $S_n = C_S n^{\frac{d}{d+2(\beta-2)}}$, $B_n = C_B n^{\frac{d}{d+2(\beta-2)}} \log n$ with $C_W, C_S, C_B$ being constants independent of $n$.

Correspondingly, we construct the entire DNN-class as

$$\mathcal{F} = \cup_{W=1}^{\infty} \cup_{S=1}^{\infty} \cup_{B>0} \mathcal{F}(L, W, S, B),$$

where $L$ is as specified above.

Below we sketch the proof to verify that the sieve $\mathcal{F}_n$ as constructed above and the prior $\Pi$ as stated in Section 2.4 satisfies the three conditions in Lemma 4.1. The detailed proofs can be found in Appendix A.3.

**Verifying (8)** First, thanks to (Lu et al., 2021a), for any $\theta_1, \theta_2 \in \Theta(L, W, S, B)$, we have

$$\sup_{x \in \Omega} |\text{clip} \circ f_{\theta_1}(x) - \text{clip} \circ f_{\theta_2}(x)|$$
$$\vee \sup_{x \in \Omega} \|\nabla(\text{clip} \circ f_{\theta_1})(x) - \nabla(\text{clip} \circ f_{\theta_2})(x)\|_{\infty}$$
$$\vee \sup_{x \in \Omega} \|\nabla^2(\text{clip} \circ f_{\theta_1}(x)) - \nabla^2(\text{clip} \circ f_{\theta_2})(x)\|_{\infty}$$
$$= O(W^{\frac{3L-1-1}{2}}(B \vee d)^{\frac{5 \cdot 3L-1-1}{2}}) \cdot \|\theta_1 - \theta_2\|_{\infty}.$$

Then using the boundedness assumption of the coefficients, it is not hard to show that

$$\sup_{x \in \Omega} |-\text{div}(A\nabla(\text{clip} \circ f_{\theta_1} - \text{clip} \circ f_{\theta_2}))(x)$$
$$+ V(x)(\text{clip} \circ f_{\theta_1} - \text{clip} \circ f_{\theta_2})(x)|$$
$$= O(W^{\frac{3L-1-1}{2}}(B \vee d)^{\frac{5 \cdot 3L-1-1}{2}}) \cdot \|\theta_1 - \theta_2\|_{\infty} \quad (12)$$
$$=: C \cdot \|\theta_1 - \theta_2\|,$$

where $C$ depends on $L, W, B$. Hence the covering number of DNN-space can be bounded by the covering number of the corresponding parameter space:

$$\log \mathcal{N}(\frac{\epsilon_n}{4}, \mathcal{F}_n, \|-\text{div}(A\nabla \cdot) + V \cdot \|_{n,1})$$
$$\lesssim \log \left(\frac{\epsilon_n}{4C|\Omega|B_n}\right)^{-S_n}$$
$$\lesssim S_n \log \left(\frac{4C|\Omega|B_n}{\epsilon_n}\right)$$
$$\lesssim n\epsilon_n^2,$$

where the last step follows from the construction of the sieve $\mathcal{F}_n$ where $W_n = C_W n^{\frac{d}{d+2(\beta-2)}}$, $S_n = C_S n^{\frac{d}{d+2(\beta-2)}}$, $B_n = C_B n^{\frac{d}{d+2(\beta-2)}} \log n$. The second entropy condition follows by an analogous argument.

**Verifying (9)** By the construction of $\mathcal{F}_n$ and Proposition 4.2, we can find a $u_n^*$ such that $\|u_n^* - u^*\|_{C^2(\bar{\Omega})} \lesssim \epsilon_n$, and $u_n^* \in \mathcal{F}(L, W_n^*, S_n^*, B_n^*) \subset \mathcal{F}_n$ with $W_n^* = C_W^* n^{\frac{d}{d+2(\beta-2)}}$, $S_n^* = C_S^* n^{\frac{d}{d+2(\beta-2)}}$, $B_n^* = C_B^* n^{\frac{d}{d+2(\beta-2)}}$. Then by a triangle inequality, we can lower bound $\Pi(u \in \mathcal{F} : \mathcal{E}_n(u; \mathcal{D}^{(n)}) \leq \epsilon_n^2)$ by the posterior mass of a subset of $\mathcal{F}_n$, each function in which shares the same architecture as $u_n^*$, denoted by $\gamma_n^*$, and is within $O(\epsilon_n)$-neighborhood of $u_n^*$. Moreover, because of (12), the mass of the subset can be further lower bounded by the a subset of functions whose parameters is within $O(\epsilon_n/C)$-neighborhood of $\theta_n^*$, where $\theta_n^*$ is the parameters of $u_n^*$. Specifically, we have

$$\Pi(u \in \mathcal{F} : \mathcal{E}_n(u; \mathcal{D}^{(n)}) \leq \epsilon_n^2)$$
$$\geq \Pi_\theta(W = W_n^*, \gamma = \gamma_n^*, 2B_n^* > B \geq B_n^*)\left(\frac{\epsilon_n}{2CB_n^*}\right)^{S_n^*}$$
$$\gtrsim \frac{1}{W_n^*!} \frac{1}{(2L(W_n^*)^2)^{S_n^*}} \exp(-\lambda B_n^*)\left(\frac{\epsilon_n}{2CB_n^*}\right)^{S_n^*}$$
$$\geq \exp(-\xi n\epsilon_n^2),$$

where the constant $\xi$ depends on $C_W^*, C_S^*, C_B^*$.

**Verifying (10)** Note that

$$\Pi(\mathcal{F}\backslash\mathcal{F}_n) \leq \Pi_\theta(W > W_n) + \Pi_\theta(W \leq W_n, S > S_n)$$
$$+ \Pi_\theta(B > B_n).$$

It is not hard to show that each term on the right hand side decays with a rate of at least $e^{-(\xi+2)n\epsilon_n^2}$ provided that the constants $C_W, C_S, C_B$ are chosen to be sufficiently large.

**Bounding the generalization gap** The proposition below bounds the generalization gap between the empirical PINN-loss and the population counterpart.

**Proposition 4.3.** *Assume $u^*$ satisfies the same assumption as in Theorem 3.1. Take $\mathcal{F}_n$ be as the same in (11). Then with probability at least $1 - 2\exp(-n\epsilon_n^2)$, we have*

$$\sup_{u \in \mathcal{F}_n}\left[\mathcal{E}(u) - \mathcal{E}_n(u; \mathcal{D}^{(n)})\right] \lesssim \epsilon_n^2.$$

The proof of Proposition 4.3 follows the standard local Rademacher complexity argument which also relies on the uniform boundedness of the DNN-class thanks to the clip operation. We present its proof in Appendix A.4.

**Proof of Theorem 3.1**    From the proof of Lemma 4.1, we have

$$\Pi(u \in \mathcal{F}_n : \| - \operatorname{div}(A\nabla u) + Vu - f\|_{n,1} \gtrsim \epsilon_n | \mathcal{D}^{(n)})$$
$$= o_p(1).$$

Proposition 4.3 implies that for all $u \in \mathcal{F}_n$,

$$\| - \operatorname{div}(A\nabla u) + Vu - f\|^2_{L^2(\Omega)}$$
$$ - \| - \operatorname{div}(A(u - u^*)) + V(u - u^*)\|^2_{n,1} \lesssim \epsilon_n^2,$$

with probability $1 - o(1)$. Combining these two bounds above yields

$$\Pi(u \in \mathcal{F}_n : \| - \operatorname{div}(A\nabla u) + Vu - f\|_{L^2(\Omega)} \gtrsim \epsilon_n | \mathcal{D}^{(n)})$$
$$= o_p(1).$$

Similarly, we have

$$\Pi(u \in \mathcal{F}_n : \lambda\|u - g\|_{L^2(\partial\Omega)} \gtrsim \epsilon_n | \mathcal{D}^{(n)}) = o_p(1).$$

Moreover, the proof of Lemma 4.1 implies that

$$\Pi(\mathcal{F} \backslash \mathcal{F}_n | \mathcal{D}^{(n)}) = o_p(1).$$

The desired result then follows by assembling the three inequalities listed above.

### 4.2. Proof sketch of Theorem 3.5

The proof idea follows from the observation that the PINN-loss $\mathcal{E}(\cdot)$ is a summation of two nonnegative terms, $\| - \operatorname{div}(A\nabla \cdot) + V \cdot - f\|^2_{L^2(\Omega)}$ and $\| \cdot - g\|^2_{L^2(\partial\Omega)}$. Hence the minimax risk can be bounded from below using the lower bounds for two separate minimax problems:

$$2\inf_{\psi} \sup_{u^*} \mathbb{E}\mathcal{E}(\psi(\mathcal{D}))$$
$$\geq \inf_{\psi} \sup_{u^*} \mathbb{E}\| - \operatorname{div}(A\nabla\psi(\mathcal{D})) + V\psi(\mathcal{D}) - f\|^2_{L^2(\Omega)}$$
$$+ \lambda \inf_{\psi} \sup_{u^*} \mathbb{E}\|\psi(\mathcal{D}) - g\|^2_{L^2(\partial\Omega)},$$

where the supremum is taken over all $u^* \in C^\beta(\bar{\Omega})$ such that $-\operatorname{div}(A\nabla u^*) + Vu^* = f$ in $\Omega$ and $u^* = g$ on $\partial\Omega$ for $f \in C^{\beta-2}(\bar{\Omega})$ and $g \in C^\beta(\partial\Omega)$, and the infimum is taken over all estimators $\psi : (\mathbb{R}^d)^{\otimes n_1} \times \mathbb{R}^{\otimes n_1} \times (\mathbb{R}^d)^{\otimes n_2} \times \mathbb{R}^{\otimes n_2} \to C^\beta(\bar{\Omega})$. The difference between the latter two minimax problems and the standard minimax theory for nonparametric regression is that the input of the estimator consists of the data both within the domain and from the boundary, i.e. $\mathcal{D} = \{X_i, f_i\}_{i=1}^{n_1} \cup \{Y_j, g_j\}_{j=1}^{n_2}$. To address the first problem, we construct a packing consisting of functions that vanish on the boundary. This ensures that the KL divergence between the distribution of $\mathcal{D}$ induced by any two distinct functions in the packing depends only on $n_1$, rather on $n_2$. Consequently, by applying local Fano's method, we

can select the packing size depending only $n_1$ to control the KL divergence, and this yields a rate $n_1^{-\frac{2(\beta-2)}{d+2(\beta-2)}}$ for the first problem. Similarly, we construct a packing so that the functions in the packing satisfy $-\operatorname{div}(A\nabla\cdot) + V\cdot = 0$ in $\Omega$. Consequently, the KL divergence between the distribution of $\mathcal{D}$ induced by any two functions in the packing depends only on $n_2$. By local Fano's method, we obtain a rate $n_2^{-\frac{2\beta}{d-1+2\beta}}$ for the second problem.

We present the detailed proof in Appendix B.3.

## 5. Conclusion and Discussion

In this paper, we prove that a Bayesian PINN equipped with a sparse prior attains an optimal posterior contraction rate for estimating the $\beta$-Hölder solution of a general class of elliptic equations from noisy observations of the source term and Dirichlet boundary data. Our results further show that the Bayesian PINN is rate-adaptive with respect to the (unknown) smoothness level of the true solution.

Our work has several limitations that open several directions for future research. First, our posterior contraction is quantified in the norm induced by the PINN-loss, which implies contraction in the $H^s$-norm only for $s \leq 1/2$ (see Remark 3.3), but not for higher $s$. By contrast, for elliptic PDEs with homogeneous Dirichlet data, a vanilla PINN with a hard zero boundary constraint attains the same optimal rate as ours in the stronger $H^2$-norm (Lu et al., 2021a). The main obstacle to extending our contraction rate to $H^2$-norm in the non-homogeneous setting is that the PINN loss uses an $L^2$-residual on the boundary which is too weak to control the $H^2$-error. Determining whether the same measurement model can yield an optimal rate in $H^2$-norm perhaps via a stronger boundary penalty or alternative variational formulations remains an interesting open problem.

Second, our lower bound allows unequal data sizes in interior and boundary measurements, suggesting that the minimax rate (in the PINN-loss norm) may be retained even with fewer boundary measurements. However, our current proof technique for the posterior contraction result (Lemma 4.1) does not directly extend to the case where the numbers of internal and boundary measurements differ. Establishing a minimax optimal posterior contraction rate in this imbalanced-measurement setting is an important direction.

Finally, while the present paper considers only Hölder smooth solutions, it will be very interesting to consider PDEs that admit irregular solutions. In fact, nonlinear adaptive wavelet methods have proven (Binev et al., 2004; Cascon et al., 2008; DeVore, 1998; 2002; Cohen et al., 2001; Dahmen & DeVore, 2002) to be *locally adaptive*, i.e. they achieve optimal rates for approximating Besov solutions of PDEs with irregular data without knowing the hetero-

geneous smoothness of solutions. Whether DNN-based approaches can realize comparable local adaptivity in such non-smooth regimes remains an open question. We plan to investigate these directions in future work.

## Acknowledgment

YL and YZ thank the support from the NSF Award DMS-2436333 and NSF CAREER Award DMS-2442463.

## Impact statement

This paper presents work whose goal is to advance the field of Machine Learning. There are many potential societal consequences of our work, none which we feel must be specifically highlighted here.

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

# A. Proof of the results in Section 4.1

## A.1. Proof of Lemma 4.1

*Proof of Lemma 3.1.* Note that for sufficiently large $M > 0$ and $J > 0$, we have

$$
\begin{aligned}
&P_{u^*}^{(n)} Q_{u^*}^{(n)} \left[ \Pi(u \in \mathcal{F} : \mathcal{E}_n(u; \mathcal{D}^{(n)}) > (\lambda+1)J^2 M^2 \epsilon_n^2 | \mathcal{D}^{(n)}) \right] \\
=&P_{u^*}^{(n)} Q_{u^*}^{(n)} \left[ \Pi(u \in \mathcal{F} : \| -\mathrm{div}(A\nabla(u-u^*)) + V(u-u^*) \|_{n,1}^2 + \lambda \|u-u^*\|_{n,2}^2 > (\lambda+1)J^2 M^2 \epsilon_n^2 | \mathcal{D}^{(n)}) \right] \\
\leq&P_{u^*}^{(n)} Q_{u^*}^{(n)} \left[ \Pi(u \in \mathcal{F}_n : \| -\mathrm{div}(A\nabla(u-u^*)) + V(u-u^*) \|_{n,1}^2 + \lambda \|u-u^*\|_{n,2}^2 > (\lambda+1)J^2 M^2 \epsilon_n^2 | \mathcal{D}^{(n)}) \right] \\
&+ P_{u^*}^{(n)} Q_{u^*}^{(n)} \left[ \Pi(\mathcal{F} \backslash \mathcal{F}_n | \mathcal{D}^{(n)}) \right] \\
\leq&P_{u^*}^{(n)} Q_{u^*}^{(n)} \left[ \Pi(u \in \mathcal{F}_n : \| -\mathrm{div}(A\nabla(u-u^*)) + V(u-u^*) \|_{n,1} > JM\epsilon_n | \mathcal{D}^{(n)}) \right] \\
&+ P_{u^*}^{(n)} Q_{u^*}^{(n)} \left[ \Pi(u \in \mathcal{F}_n : \|u-u^*\|_{n,2} > JM\epsilon_n | \mathcal{D}^{(n)}) \right] + P_{u^*}^{(n)} Q_{u^*}^{(n)} \left[ \Pi(\mathcal{F} \backslash \mathcal{F}_n | \mathcal{D}^{(n)}) \right].
\end{aligned}
\tag{13}
$$

The proof can be structured as follows: the first two terms can be handled by the conditions (8) and (9), and the last term can be bounded by using the conditions (9) and (10).

By the condition (8) and Lemma C.1 and C.2, there exists a test $\phi_1$ on $\{X_i, f_i\}_{i=1}^n$ such that

$$
P_{u^*}^{(n)} \phi_1 \leq e^{Cn\epsilon_n^2} \frac{e^{-\frac{1}{32}nM^2\epsilon_n^2}}{1 - e^{-\frac{1}{32}nM^2\epsilon_n^2}}, \quad P_u^{(n)}(1 - \phi_1) \leq e^{-\frac{1}{32}nM^2\epsilon_n^2 j^2},
\tag{14}
$$

for all $j \in \mathbb{N}$ and $u \in \mathcal{F}$ with $\| -\mathrm{div}(A\nabla(u-u^*)) + V(u-u^*) \|_{n,1} > jM\epsilon_n$. Then by the first inequality of (14), we have

$$
\begin{aligned}
&P_{u^*}^{(n)} Q_{u^*}^{(n)} \left[ \Pi(u \in \mathcal{F}_n : \| -\mathrm{div}(A\nabla(u-u^*)) + V(u-u^*) \|_{n,1} > JM\epsilon_n | \mathcal{D}^{(n)})\phi_1 \right] \\
\leq&P_{u^*}^{(n)} Q_{u^*}^{(n)}(\phi_1) = P_{u^*}^{(n)}(\phi_1) \leq e^{Cn\epsilon_n^2} \frac{e^{-\frac{1}{32}nM^2\epsilon_n^2}}{1 - e^{-\frac{1}{32}nM^2\epsilon_n^2}}.
\end{aligned}
\tag{15}
$$

Moreover, by Fubini's theorem and the second inequality of (14),

$$
\begin{aligned}
&P_{u^*}^{(n)} Q_{u^*}^{(n)} \left[ \int_{\| -\mathrm{div}(A\nabla(u-u^*)) + V(u-u^*) \|_{n,1} \geq JM\epsilon_n} \frac{p_u^{(n)}}{p_{u^*}^{(n)}} \frac{q_u^{(n)}}{q_{u^*}^{(n)}} d\Pi(u)(1-\phi_1) \right] \\
=& \int_{\| -\mathrm{div}(A\nabla(u-u^*)) + V(u-u^*) \|_{n,1} \geq JM\epsilon_n} P_u^{(n)} Q_u^{(n)} \left[(1-\phi_1)\right] d\Pi(u) \leq e^{-\frac{1}{36}nM^2\epsilon_n^2 J^2}.
\end{aligned}
\tag{16}
$$

Now together with the prior condition (9), we can apply Lemma C.3 to lower bound the normalization constant such that for any $C_1 > 0$

$$
\begin{aligned}
\int \frac{p_u^{(n)}}{p_{u^*}^{(n)}} \frac{q_u^{(n)}}{q_{u^*}^{(n)}} d\Pi(u) &> \int_{\mathcal{E}_n(u;\mathcal{D}^{(n)}) \leq \epsilon_n^2} \frac{p_u^{(n)}}{p_{u^*}^{(n)}} \frac{q_u^{(n)}}{q_{u^*}^{(n)}} d\Pi(u) \\
&> \Pi(u \in \mathcal{F} : \mathcal{E}_n(u; \mathcal{D}^{(n)}) \leq \epsilon_n^2) e^{-(1+C_1)n\epsilon_n^2} \\
&> e^{-\xi n\epsilon_n^2 - (1+C_1)n\epsilon_n^2},
\end{aligned}
\tag{17}
$$

with probability at least $1 - C_1^{-1}(n\epsilon_n^2)^{-1}$. We denote the event on which (17) holds as $E$. By combining (16) and (17), we reach

$$
\begin{aligned}
&P_{u^*}^{(n)} Q_{u^*}^{(n)} \left[ \Pi(u \in \mathcal{F}_n : \| -\mathrm{div}(A\nabla(u-u^*)) + V(u-u^*) \|_{n,1} > JM\epsilon_n | \mathcal{D}^{(n)})(1-\phi_1) \right] \\
\leq&P_{u^*}^{(n)} Q_{u^*}^{(n)} \left[ \Pi(u \in \mathcal{F}_n : \| -\mathrm{div}(A\nabla(u-u^*)) + V(u-u^*) \|_{n,1} > JM\epsilon_n | \mathcal{D}^{(n)})(1-\phi_1)\mathbf{1}_E \right] + P_{u^*}^{(n)} Q_{u^*}^{(n)}(\mathbf{1}_{E^c}) \\
=&P_{u^*}^{(n)} Q_{u^*}^{(n)} \left[ \frac{\int_{\| -\mathrm{div}(A\nabla(u-u^*)) + V(u-u^*) \|_{n,1} > JM\epsilon_n} \frac{p_u^{(n)}}{p_{u^*}^{(n)}} \frac{q_u^{(n)}}{q_{u^*}^{(n)}} d\Pi(u)}{\int \frac{p_u^{(n)}}{p_{u^*}^{(n)}} \frac{q_u^{(n)}}{q_{u^*}^{(n)}} d\Pi(u)} (1-\phi_1)\mathbf{1}_E \right] + P_{u^*}^{(n)} Q_{u^*}^{(n)}(\mathbf{1}_{E^c}) \\
\leq&e^{-\frac{1}{36}nM^2\epsilon_n^2 J^2 + \xi n\epsilon_n^2 + (1+C_1)n\epsilon_n^2} + C_1^{-1}(n\epsilon_n^2)^{-1}.
\end{aligned}
\tag{18}
$$

Consequently, combining (15) and (18) yields a bound on the first term in (13)

$$P_{u^*}^{(n)} Q_{u^*}^{(n)} \left[ \Pi(u \in \mathcal{F}_n : \| - \mathrm{div}(A\nabla(u - u^*)) + V(u - u^*) \|_{n,1} > JM\epsilon_n | \mathcal{D}^{(n)}) \right]$$

$$\leq e^{Cn\epsilon_n^2} \frac{e^{-\frac{1}{32} nM^2 \epsilon_n^2}}{1 - e^{-\frac{1}{32} nM^2 \epsilon_n^2}} + e^{-\frac{1}{36} nM^2 \epsilon_n^2 J^2 + \xi n\epsilon_n^2 + (1+C_1)n\epsilon_n^2} + C_1^{-1}(n\epsilon_n^2)^{-1}. \tag{19}$$

Analogously, the second term in (13) can be bounded by

$$P_{u^*}^{(n)} Q_{u^*}^{(n)} \left[ \Pi(u \in \mathcal{F}_n : \|u - u^*\|_{n,2} > JM\epsilon_n | \mathcal{D}^{(n)}) \right]$$

$$\leq e^{Cn\epsilon_n^2} \frac{e^{-\frac{1}{32} nM^2 \epsilon_n^2}}{1 - e^{-\frac{1}{32} nM^2 \epsilon_n^2}} + e^{-\frac{1}{36} nM^2 \epsilon_n^2 J^2 + \xi n\epsilon_n^2 + (1+C_1)n\epsilon_n^2} + C_1^{-1}(n\epsilon_n^2)^{-1}. \tag{20}$$

To bound the third term in (13), note that by Fubini's theorem and the sieve condition (10),

$$P_{u^*}^{(n)} Q_{u^*}^{(n)} \left[ \int_{\mathcal{F} \backslash \mathcal{F}_n} \frac{p_u^{(n)} q_u^{(n)}}{p_{u^*}^{(n)} q_{u^*}^{(n)}} d\Pi(u) \right] = \left[ \int_{\mathcal{F} \backslash \mathcal{F}_n} P_u^{(n)} Q_u^{(n)} d\Pi(u) \right] \leq \Pi(\mathcal{F} \backslash \mathcal{F}_n) = o(e^{-(\xi+2)n\epsilon_n^2}). \tag{21}$$

Then by (17) and (21), we have

$$P_{u^*}^{(n)} Q_{u^*}^{(n)} \left[ \Pi(\mathcal{F} \backslash \mathcal{F}_n | \mathcal{D}^{(n)}) \right]$$

$$\leq P_{u^*}^{(n)} Q_{u^*}^{(n)} \left[ \Pi(\mathcal{F} \backslash \mathcal{F}_n | \mathcal{D}^{(n)}) \mathbf{1}_E \right] + P_{u^*}^{(n)} Q_{u^*}^{(n)} (\mathbf{1}_{E^c})$$

$$\leq P_{u^*}^{(n)} Q_{u^*}^{(n)} \left[ \frac{\int_{\mathcal{F} \backslash \mathcal{F}_n} \frac{p_u^{(n)} q_u^{(n)}}{p_{u^*}^{(n)} q_{u^*}^{(n)}} d\Pi(u)}{\int \frac{p_u^{(n)} q_u^{(n)}}{p_{u^*}^{(n)} q_{u^*}^{(n)}} d\Pi(u)} \right] + P_{u^*}^{(n)} Q_{u^*}^{(n)} (\mathbf{1}_{E^c}) \tag{22}$$

$$\leq e^{-(\xi+2)n\epsilon_n^2 + \xi n\epsilon_n^2 + (1+C_1)n\epsilon_n^2} o(1) + C_1^{-1}(n\epsilon_n^2)^{-1} = e^{(C_1 - 1)n\epsilon_n^2} o(1) + C_1^{-1}(n\epsilon_n^2)^{-1}.$$

Finally, combining (19), (20) and (22) yields an upper bound on (13)

$$P_{u^*}^{(n)} Q_{u^*}^{(n)} \left[ \Pi(u \in \mathcal{F} : \mathcal{E}_n(u; \mathcal{D}^{(n)}) > (\lambda + 1) J^2 M^2 \epsilon_n^2 | \mathcal{D}^{(n)}) \right]$$

$$\leq 2e^{Cn\epsilon_n^2} \frac{e^{-\frac{1}{32} nM^2 \epsilon_n^2}}{1 - e^{-\frac{1}{32} nM^2 \epsilon_n^2}} + 2e^{-\frac{1}{36} nM^2 \epsilon_n^2 J^2 + \xi n\epsilon_n^2 + (1+C_1)n\epsilon_n^2} + 3C_1^{-1}(n\epsilon_n^2)^{-1} + e^{(C_1 - 1)n\epsilon_n^2} o(1),$$

and by taking taking $M, J$ to be sufficiently large and $C_1 = 1$, the upper bound tends to zero as $n \to \infty$. $\qquad \square$

### A.2. Proof of Proposition 4.2

**Univariate B-splines** Fix an arbitrary integer $l \in \mathbb{N}$. Consider a corresponding uniform partition $\pi_l$ of $[0, 1]$:

$$\pi_l : 0 = t_0^{(l)} < t_1^{(l)} < \cdots < t_{l-1}^{(l)} < t_l^{(l)} = 1,$$

where $t_i^{(l)} = \frac{i}{l}$ for all $0 \leq i \leq l$. For any $k \in \mathbb{N}$, we define an extended partition $\pi_{l,k}$ as

$$\pi_{l,k} : t_{-k+1}^{(l)} = \cdots = t_{-1}^{(l)} = 0 = t_0^{(l)} < t_1^{(l)} < \cdots < t_{l-1}^{(l)} < t_l^{(l)} = 1 = t_{l+1}^{(l)} = \cdots = t_{l+k-1}^{(l)}.$$

Based on the extended partition $\pi_{l,k}$, the univariate B-splines of order $k$ are defined by

$$N_{l,i}^{(k)}(x) := (-1)^k (t_{i+k}^{(l)} - t_i^{(l)}) \cdot [t_i^{(l)}, \ldots, t_{i+k}^{(l)}] (\max\{(x - t), 0\})^{k-1}, \quad x \in [0, 1], \ i \in I_{l,k},$$

where $I_{l,k} = \{-k+1, -k+2, \ldots, l-1\}$ and $[t_i^{(l)}, \ldots, t_{i+k}^{(l)}]$ denotes the divided difference operator. Equivalently, for any $x \in [0, 1]$, the univariate B-splines $N_{l,i}^{(k)}(x)$ can be expressed explicitly as

$$N_{l,i}^{(k)}(x) = \frac{l^{k-1}}{(k-1)!} \sum_{j=0}^{k} (-1)^j \binom{k}{j} \left( \max \left\{ x - \frac{(i+j)}{l}, 0 \right\} \right)^{k-1}, \quad i \in I_{l,k}. \tag{23}$$

**Multivariate B-splines**  For any index vector $\boldsymbol{i} = (i_1, i_2, \ldots, i_d) \in I_{l,k}^d$, the corresponding multivariate B-spline is defined as

$$N_{l,\boldsymbol{i}}^{(k)}(x) := \prod_{j=1}^{d} N_{l,i_j}^{(k)}(x_j).$$

**Lemma A.1** ((Schumaker, 2007)).  *Consider a partition of $[0,1]^d$,*

$$\Omega_{\boldsymbol{i}} = \{x \in [0,1]^d : x_j \in [t_{i_j}, t_{i_j+k}], \ 1 \leq j \leq d\}.$$

*There exists a dual basis $\{\lambda_{\boldsymbol{i}}\}_{i \in I_{k,l}^d}$ satisfying $\lambda_{\boldsymbol{j}} N_{l,\boldsymbol{i}}^{(k)} = \delta_{\boldsymbol{i},\boldsymbol{j}}$. Moreover, for any $p \in [1, \infty]$,*

$$|\lambda_{\boldsymbol{i}}(f)| \leq 9^{d(k-1)}(2k+1)^d \left(\frac{k}{l}\right)^{-d/p} \|f\|_{L^p(\Omega_{\boldsymbol{i}})}, \quad f \in L^p([0,1]^d). \tag{24}$$

*Define the corresponding interpolation operator $Q_{k,l}$ as*

$$Q_{k,l}f := \sum_{\boldsymbol{i} \in I_{k,l}^d} \lambda_{\boldsymbol{i}}(f) N_{l,\boldsymbol{i}}^{(k)}, \quad f \in L^p([0,1]^d). \tag{25}$$

*Moreover, let $0 < m < \beta \leq k$. Then for any $f \in C^\beta([0,1]^d)$, we have*

$$\|f - Q_{k,l}f\|_{C^m([0,1]^d)} \lesssim l^{-(\beta-m)} \|f\|_{C^\beta([0,1]^d)}.$$

*Proof of Proposition 4.2.* Take $D > 0$ to be such that $\Omega \subset [-D, D]^d$. By Whitney extension theorem, there exists an extension $\bar{f} \in C^\beta([-D,D]^d)$ of $f \in C^\beta(\bar{\Omega})$ such that $\bar{f}|_\Omega = f$ and $\|\bar{f}\|_{C^\beta([-D,D]^d)} \leq C_D \|f\|_{C^\beta(\bar{\Omega})}$ where $C_D$ is some constant not depending on $f$. Define $\tilde{f}(x) = \bar{f}(2Dx - D)$ so that $\tilde{f} \in C^\beta([0,1]^d)$.

Take $k \geq \beta$ such that $k - 1$ is a multiple of 3. Then by Lemma A.1, there exists $Q_{k,l}$ such that

$$\|\tilde{f} - Q_{k,l}\tilde{f}\|_{C^2([0,1]^d)} \lesssim l^{-(\beta-2)}\|\tilde{f}\|_{C^\beta([0,1]^d)} \lesssim l^{-(\beta-2)}\|\bar{f}\|_{C^\beta([-D,D]^d)} \lesssim l^{-(\beta-2)}\|f\|_{C^\beta(\bar{\Omega})},$$

where the last inequality follows from the boundedness of the extension operator. Define $\phi(x) = [Q_{k,l}\tilde{f}](\frac{x+D}{2D})$ so that we have

$$\|f - \phi\|_{C^2(\Omega)} \leq \|\bar{f} - \phi\|_{C^2([-D,D]^d)} \lesssim \|\tilde{f} - Q_{k,l}\tilde{f}\|_{C^2([0,1]^d)} \lesssim l^{-(\beta-2)}\|f\|_{C^\beta(\Omega)}.$$

We now need to construct a $\sigma_3$-network to implement $Q_{k,l}\tilde{f}(\frac{x+D}{2D})$, the explicit formula of which can be found in (25).

Thanks to Lemma C.4, we can implement $x \mapsto \frac{x+D}{2D}$ for $x \in [-D, D]$ by a single layer $\sigma_3$-network.

To implement $Q_{k,l}\tilde{f}$, we first construct a neural network to implement $(\max\{x - c, 0\})^{k-1}$ as follows. The first layer prepares $\frac{k-1}{3}$ copies of $\sigma_3(x - c)$ and $2^{\lceil \log_2 \frac{k-1}{3} \rceil} - \frac{k-1}{3}$ copies of 1, which requires $2^{\lceil \log_2 \frac{k-1}{3} \rceil}$ neurons. The second layer performs multiplication between adjacent neurons, which requires 9 neurons for each pair according to Lemma C.5. The total number of neuron requires in the second layer should be $9 \cdot 2^{\lceil \log_2 \frac{k-1}{3} \rceil - 1}$. Looping this procedure results in a $1 + \lceil \log_2 \frac{k-1}{3} \rceil$ layer network implementing $(\max\{x - c, 0\})^{k-1}$ precisely. Counting the number of neurons in each layer yields that the width is at most $9 \cdot 2^{\lceil \log_2 \frac{k-1}{3} \rceil - 1} = O(k)$. The number of nonzero parameters in the first layer is

$$2 \cdot \frac{k-1}{3} + 2^{\lceil \log_2 \frac{k-1}{3} \rceil} - \frac{k-1}{3},$$

and the number of nonzero parameters in the second layer is at most

$$18 \cdot 2^{\lceil \log_2 \frac{k-1}{3} \rceil - 1}.$$

From the third layer, the number of nonzero parameters in the second layers is at most

$$C2^{\lceil \log_2 \frac{k-1}{3} \rceil - l}, \quad l = 2, \ldots, \lceil \log_2 \frac{k-1}{3} \rceil,$$

where $C$ is a universal constant. Hence the sparsity can bounded by

$$\frac{k-1}{3} + C \sum_{l=0}^{\lceil \log_2 \frac{k-1}{3} \rceil} 2^l = O(k + 2^{\lceil \log_2 \frac{k-1}{3} \rceil}) = O(k).$$

Next we aim to find an efficient way to implement the coefficients of those $(\max\{x - c, 0\})^{k-1}$. A closer look into (23) shows that the coefficients can be bounded as

$$\frac{l^{k-1}}{(k-1)!} \binom{k}{j} \leq l^{k-1},$$

Then $\frac{l^{k-1}}{(k-1)!} \binom{k}{j}$ can be written as a multiplication of at most $\lceil \frac{k-1}{d} \rceil$ factors with each less than or equal to $l^d$. By the multiplication procedure displayed above, the coefficients can be implemented with $O(\log k)$ layers, $O(k)$-width and $O(k)$ nonzero parameters. So far, we can construct a $\sigma_3$-network with $L = O(\log k)$, $W = O(k)$, $S = O(k)$, $B = O(l^d)$ and it can implement each $N_{l,i}^{(k)}$ precisely.

To obtain each basis function $N_{l,i}^k$, we need to tensorize $\{N_{l,i_j}^{(k)}\}_{j=1}^d$ together, and that requires a network with $L = O(\log d + \log k)$, $W = O(dk)$, $S = O(dk)$ and $B = O(l^d)$.

Finally, we can parallelize the previous implementation for each $N_{l,i}^{(k)}$ together to obtain a network with $L = O(\log d + \log k)$, $W = O(kd(l+k)^d) = O(kd \cdot l^d)$, $S = O(kd(l+k)^d) = O(kd \cdot l^d)$ and it can implement $Q_{k,l}f := \sum_{i \in I_{k,l}^d} \lambda_i(f) N_{l,i}^{(k)}$ precisely. The boundedness $B$ can be identified from (24) and our construction above

$$B = \max\{O(l^d), |\lambda_i f|\} = \max\{O(l^d), O(9^{kd} k^d l^d)\} = O(l^d).$$

$\square$

### A.3. Verifying the conditions in Lemma 4.1

We choose the cutoff function to satisfy $\text{clip}(x) = x$ when $-K - 1 \leq x \leq K + 1$, and $\text{clip}(x) = 2(K + 1)$ when $x \geq 2(K + 1)$, and $\text{clip}(x) = -2(K + 1)$ when $x \leq -2(K + 1)$. The detailed construction can be found in Appendix D.

**Verifying (8)**  We want to show that there exists some constant $C$ such that

$$\log \mathcal{N}(\frac{\epsilon_n}{4}, \mathcal{F}_n, \| - \text{div}(A\nabla \cdot) + V \cdot \|_{n,1}) \leq Cn\epsilon_n^2,$$

$$\log \mathcal{N}(\frac{\epsilon_n}{4}, \mathcal{F}_n, \| \cdot \|_{n,2}) \leq Cn\epsilon_n^2.$$

By the construction of $\mathcal{F}_n$, we have

$$\mathcal{N}(\frac{\epsilon_n}{4}, \mathcal{F}_n, \| - \text{div}(A\nabla \cdot) + V \cdot \|_{n,1}) \leq \sum_{W=1}^{W_n} \sum_{S=1}^{S_n} \mathcal{N}(\frac{\epsilon_n}{4}, \mathcal{F}(L, W, S, B_n), \| - \text{div}(A\nabla \cdot) + V \cdot \|_{n,1})$$

$$\leq W_n S_n \mathcal{N}(\frac{\epsilon_n}{4}, \mathcal{F}(L, W_n, S_n, B_n), \| - \text{div}(A\nabla \cdot) + V \cdot \|_{n,1}). \tag{26}$$

Lemma C.6 shows that for any $\theta_1, \theta_2 \in \Theta(L, W, S, B)$, we have

$$\sup_{x \in \Omega} |\text{clip} \circ f_{\theta_1}(x) - \text{clip} \circ f_{\theta_2}(x)| \vee \sup_{x \in \Omega} \|\nabla(\text{clip} \circ f_{\theta_1})(x) - \nabla(\text{clip} \circ f_{\theta_2})(x)\|_\infty$$

$$\vee \sup_{x \in \Omega} \|\nabla^2(\text{clip} \circ f_{\theta_1})(x) - \nabla^2(\text{clip} \circ f_{\theta_2})(x)\|_\infty \tag{27}$$

$$\leq C_0 W^{\frac{3^L - 1}{2}} (B \vee d)^{\frac{5 \cdot 3^L - 1}{2}} \cdot \|\theta_1 - \theta_2\|_\infty =: C_p \cdot \|\theta_1 - \theta_2\|_\infty$$

By the boundedness assumption on the coefficients $A$ and $V$, we further have

$$
\begin{aligned}
&\sup_{x\in\Omega} |-\operatorname{div}(A\nabla(\operatorname{clip}\circ f_{\theta_1} - \operatorname{clip}\circ f_{\theta_2}))(x) + V(x)(\operatorname{clip}\circ f_{\theta_1} - \operatorname{clip}\circ f_{\theta_2})(x)| \\
&\leq \sup_{x\in\Omega} |A(x)\cdot\nabla^2(\operatorname{clip}\circ f_{\theta_1} - \operatorname{clip}\circ f_{\theta_2})(x)| + \sup_{x\in\Omega} |\nabla A(x)\cdot\nabla(\operatorname{clip}\circ f_{\theta_1} - \operatorname{clip}\circ f_{\theta_2})(x)| \\
&\quad + \sup_{x\in\Omega} |V(x)(\operatorname{clip}\circ f_{\theta_1} - \operatorname{clip}\circ f_{\theta_2})(x)| \\
&\leq (C_A d^2 + C_A d + V_{\max})C_p\|\theta_1-\theta_2\|_\infty =: C_c C_p\|\theta_1-\theta_2\|_\infty.
\end{aligned}
\tag{28}
$$

Then we can bound the covering number of the function space by the covering number of the parameter space

$$
\begin{aligned}
\mathcal{N}(\frac{\epsilon_n}{4}, \mathcal{F}(L,W,S,B), \|-\operatorname{div}(A\nabla\cdot)+V\cdot\|_{n,1}) &\leq \binom{T}{S}\left(\frac{\epsilon_n/4}{C_c C_p|\Omega|B}\right)^{-S} \\
&\leq \binom{2LW^2}{S}\left(\frac{\epsilon_n/4}{C_c C_0|\Omega|W^{\frac{3L-1}{2}}(B\vee d)^{\frac{5\cdot 3^{L}-1}{2}}B}\right)^{-S} \\
&\leq (2LW^2)^S\left(\frac{\epsilon_n/4}{C_c C_0|\Omega|W^{\frac{3L-1}{2}}(B\vee d)^{\frac{5\cdot 3^{L}-1}{2}}B}\right)^{-S}.
\end{aligned}
\tag{29}
$$

Combining (26) and (29) yields

$$
\begin{aligned}
&\log\mathcal{N}(\frac{\epsilon_n}{4}, \mathcal{F}_n, \|-\operatorname{div}(A\nabla\cdot)+V\cdot\|_{n,1}) \\
&\leq \log W_n + \log S_n + S_n\log(2LW_n^2) - S_n\log\left(\frac{\epsilon_n/4}{C_c C_0|\Omega|W_n^{\frac{3L-1}{2}}(B_n\vee d)^{\frac{5\cdot 3^{L}-1}{2}}B_n}\right) \\
&\leq C_1 n^{\frac{d}{d+2(\beta-2)}}\log n = C_1 n\epsilon_n^2,
\end{aligned}
$$

where the last step follows from the construction of $\mathcal{F}_n$ that $W_n = C_W n^{\frac{d}{d+2(\beta-2)}}, S_n = C_S n^{\frac{d}{d+2(\beta-2)}}, B_n = C_B n^{\frac{d}{d+2(\beta-2)}}\log n$, and $C_1$ depends on $L, C_W, C_S, C_B, C_0, C_A, V_{\max}, \Omega$.

Similarly, one can obtain

$$
\sup_{\epsilon>\epsilon_n}\mathcal{N}(\frac{\epsilon}{4}, \mathcal{F}_n, \|\cdot\|_{n,2}) \leq C_2 n\epsilon_n^2,
$$

where $C_2$ depends on $L, C_W, C_S, C_B, C_0, \Omega$. Then we can choose $C = \max\{C_1, C_2\}$.

**Verifying 9** We want to show that there exists some $\xi > 0$ such that

$$
\Pi(u\in\mathcal{F}: \mathcal{E}_n(u;\mathcal{D}^{(n)}) \leq \epsilon_n^2) \geq \exp(-\xi n\epsilon_n^2).
$$

Then by Proposition 4.2, there exists a $u_n^* \in \mathcal{F}(L, W_n^*, S_n^*, B_n^*)$ with $W_n^* = C_W^* n^{\frac{d}{d+2(\beta-2)}}, S_n^* = C_S^* n^{\frac{d}{d+2(\beta-2)}}, B_n^* = C_B^* n^{\frac{d}{d+2(\beta-2)}}$ such that

$$
\|u^* - u_n^*\|_{C^2(\bar\Omega)}^2 \leq \frac{3\epsilon_n^2}{4(|\Omega|+\lambda|\partial\Omega|)}.
$$

Moreover, let $\gamma_n^*$ and $\theta_n^*$ be the architecture and the parameters of $u_n^*$ respectively, and we have

$$
\begin{aligned}
&\{u\in\mathcal{F}: \mathcal{E}_n(u;\mathcal{D}^{(n)}) \leq \epsilon_n^2\} = \{u\in\mathcal{F}: \|-\operatorname{div}(A\nabla(u-u^*))+V(u-u^*)\|_{n,1}^2 + \lambda\|u-u^*\|_{n,2}^2 \leq \epsilon_n^2\} \\
&\supset \{u\in\cup_{B\geq B_n^*}^{2B_n^*}\mathcal{F}(L,W_n^*,S_n^*,B): \|-\operatorname{div}(A\nabla(u-u^*))+V(u-u^*)\|_{n,1}^2 + \lambda\|u-u^*\|_{n,2}^2 \leq \epsilon_n^2\} \\
&\supset \{u_\theta\in\cup_{B\geq B_n^*}^{2B_n^*}\mathcal{F}(L,W_n^*,S_n^*,B): \operatorname{supp}(\theta)=\gamma_n^*, \|-\operatorname{div}(A\nabla(u-u^*))+V(u-u^*)\|_{n,1}^2 + \lambda\|u-u_n^*\|_{n,2}^2 \leq \frac{\epsilon_n^2}{4}\} \\
&\supset \{u_\theta\in\cup_{B\geq B_n^*}^{2B_n^*}\mathcal{F}(L,W_n^*,S_n^*,B): \operatorname{supp}(\theta)=\gamma_n^*, \|\theta-\theta_n^*\|_\infty \leq \frac{\epsilon_n}{C_p\sqrt{C_c^2|\Omega|+\lambda|\partial\Omega|}}\} \\
&=: \{u_\theta\in\cup_{B\geq B_n^*}^{2B_n^*}\mathcal{F}(L,W_n^*,S_n^*,B): \operatorname{supp}(\theta)=\gamma_n^*, \|\theta-\theta_n^*\|_\infty \leq \frac{\epsilon_n}{2C_p'}\},
\end{aligned}
$$

where the constants $C_p$ and $C_c$ are defined in (27) and (28) respectively. Let $T \asymp (W_n^*)^2$ be total number of parameters in $\mathcal{F}(L, W_n^*, S_n^*, B_n^*)$. We can lower bound the prior mass as

$$
\begin{aligned}
&\Pi(u \in \mathcal{F} : \mathcal{E}_n(u; \mathcal{D}^{(n)}) \leq \epsilon_n^2)\\
&\geq \Pi(u_\theta \in \cup_{B \geq B_n^*}^{2B_n^*} \mathcal{F}(L, W_n^*, S_n^*, B) : \mathrm{supp}(\theta) = \gamma_n^*, \|\theta - \theta_n^*\|_\infty \leq \frac{\epsilon_n}{2C_p'})\\
&\geq \Pi_\theta(W = W_n^*, \gamma = \gamma_n^*, 2B_n^* > B \geq B_n^*) \left(\frac{\epsilon_n}{4B_n^* C_p'}\right)^{S_n^*}\\
&= \frac{\lambda_W^{W_n^*}}{(e^{\lambda_W} - 1)W_n^*!} \frac{T^{\lambda_S(T - S_n^*)}}{(1 + T^{\lambda_S})^T} \left(e^{-\lambda_B B_n^*} - e^{-2\lambda_B B_n^*}\right) \left(\frac{\epsilon_n}{4B_n^* C_p'}\right)^{S_n^*}\\
&= \frac{\lambda_W^{W_n^*}}{(e^{\lambda_W} - 1)W_n^*!} \frac{1}{T^{\lambda_S S_n^*}} \exp\left(T \log\left(1 - \frac{1}{1 + T^{\lambda_S}}\right)\right) \left(e^{-\lambda_B B_n^*} - e^{-2\lambda_B B_n^*}\right) \left(\frac{\epsilon_n}{4B_n^* C_p'}\right)^{S_n^*}\\
&\geq \frac{\lambda_W^{W_n^*}}{(e^{\lambda_W} - 1)W_n^*!} \frac{1}{T^{\lambda_S S_n^*}} \exp\left(-\frac{2T}{1 + T^{\lambda_S}}\right) \frac{e^{-\lambda_B B_n^*}}{2} \left(\frac{\epsilon_n}{4B_n^* C_p'}\right)^{S_n^*}\\
&= \exp\left(W_n^* \log \lambda_W - \lambda_W - \log(W_n^*!) - \lambda_S S_{n_1}^* \log T - \frac{2T}{1 + T^{\lambda_S}} - \lambda_B B_n^* - \log 2 + S_n^* \log\left(\frac{\epsilon_n}{4B_n^* C_p'}\right)\right)\\
&\geq \exp\left(W_n^* \log \lambda_W - W_n^* \log W_n^* - \lambda_S S_n^* \log T - \lambda_B B_n^* - \log 2 + S_n^* \log\left(\frac{\epsilon_n}{4B_n^* C_p'}\right) - \lambda_W - \frac{2T}{1 + T^{\lambda_S}}\right)\\
&\geq \exp\left(-\xi n_1^{\frac{d}{d + 2(\beta - 2)}} \log n\right),
\end{aligned}
$$

where the first inequality follows from the fact that $\log(1 - x) \geq -2x$ for $0 < x < \frac{1}{2}$, and the third inequality is a consequence of $x! \leq x^x$, and we choose $\lambda_S = 2$ such that $\frac{2T}{1 + T^{\lambda_S}} \leq 1$, and choose $\lambda_W = \lambda_B = 1$ so that $\xi$ can be selected as

$$
\xi = \frac{d}{d + 2(\beta - 2)} C_W^* + \frac{2d}{d + 2(\beta - 2)} \lambda_S C_{S,1}^* + \frac{d}{d + 2(\beta - 2)} \lambda_B C_B^* + 1.
$$

**Verifying** (10)    We want to verify that

$$
\Pi(\mathcal{F}\backslash\mathcal{F}_n) = o(\exp(-(\xi + 2)n\epsilon_n^2)).
$$

First note that

$$
\Pi(\mathcal{F}\backslash\mathcal{F}_n) \leq \Pi_\theta(W > W_n) + \Pi_\theta(W \leq W_n, S > S_n) + \Pi_\theta(B > B_n).
$$

By a Chernoff bound

$$
\Pi_\theta(W > W_n) \leq e^{-t(W_n + 1)} \mathbb{E}[e^{tW}] \propto e^{-t(W_n + 1)} e^{e^t \lambda_W - 1}.
$$

Take $t = \log W_n$ so that we have

$$
\begin{aligned}
\Pi_\theta(W > W_n) e^{(\xi + 2)n\epsilon_n^2} &\lesssim \exp\left(-W_n \log W_n + \lambda_W W_n + (\xi + 2)n\epsilon_n^2\right)\\
&\lesssim \exp\left((-\frac{d}{d + 2(\beta - 2)} C_W + 1 + \xi + 2)n\epsilon_n^2\right).
\end{aligned}
$$

Now we look at the second term

$$\Pi_\theta(W \le W_n, S > S_n) = \sum_{w=1}^{W_n} \Pi_\theta(W = w)\Pi_\theta(S > S_n | W = w)$$

$$\le \sum_{w=1}^{W_n} \Pi_\theta(W = w)\Pi_\theta(S > S_n | W = W_n) \le \Pi_\theta(S > S_n | W = W_n)$$

$$= \sum_{S>S_n}^{T} \binom{T}{S} \left(\frac{1}{1+T^{\lambda_S}}\right)^S \left(1 - \frac{1}{1+T^{\lambda_S}}\right)^{T-S} \le \sum_{S>S_n}^{T} \binom{T}{S} \left(\frac{1}{T^{\lambda_S}}\right)^S$$

$$\le \sum_{S>S_n}^{T} T^S \left(\frac{1}{T^{\lambda_S}}\right)^S \le \sum_{S>S_n} T^{-(\lambda_S-1)S} \le 2T^{-(\lambda_S-1)(S_n+1)},$$

where $T \asymp W_n^2$. Therefore we have

$$\Pi_\theta(W \le W_n, S > S_n)e^{(\xi+2)n\epsilon_n^2} \le \exp\big(-(\lambda_S - 1)(S_n + 1)\log(T) + \log 2 + (\xi + 2)n\epsilon_n^2\big)$$

$$\lesssim \exp\left(\left(-C_S \frac{d}{d + 2(\beta - 2)} + \xi + 2\right)n\epsilon_n^2\right).$$

Finally we deal with the third term,

$$\Pi_\theta(B > B_n)e^{(\xi+2)n\epsilon_n^2} = \exp\left(-\lambda_B B_n + (\xi + 2)n\epsilon_n^2\right) = \exp\left(-(C_B - \xi - 2)n\epsilon_n^2\right).$$

Then we can choose $C_W, C_S, C_B$ to be large enough so that the condition is satisfied.

### A.4. Proof of Proposition 4.3

**Lemma A.2.** *Consider a Deep neural network space $\mathcal{F}^{(1)} = \cup_{W=1}^{W_N} \cup_{S=1}^{S_N} \mathcal{F}(L, W, S, B_N)$ on the domain $\Omega$ with $L = O(1)$, $W_N = O(N)$, $S_N = O(N)$, $B_N = O(N \log N)$, where $N \in \mathbb{N}$ is fixed to be sufficiently large. For any $\rho > 0$, consider localized sets $\mathcal{M}_\rho$ defined by*

$$\mathcal{M}_\rho^{(1)} = \{u \in \mathcal{F}^{(1)} : \|u - u^*\|_{H^2(\Omega)}^2 \le \rho\},$$

*and $\mathcal{S}_\rho^{(1)}$ defined by*

$$\mathcal{S}_\rho^{(1)} = \{h = |\Omega| \cdot (-\mathrm{div}(A\nabla u) + Vu - f)^2 : u \in \mathcal{M}_\rho^{(1)}\}.$$

*Further, assume the following uniform boundedness condition*

$$\max\left\{\sup_{u\in\mathcal{F}^{(1)}} \|\nabla^2 u\|_{L^\infty(\Omega),\infty}, \sup_{u\in\mathcal{F}^{(1)}} \|\nabla u\|_{L^\infty(\Omega),\infty}, \sup_{u\in\mathcal{F}^{(1)}} \|u\|_{L^\infty(\Omega)}, \|f\|_{L^\infty(\Omega)}\right\} \le C.$$

*Then for any $\rho \gtrsim n^{-2}$, the Rademacher complexity of $\mathcal{S}_\rho^{(1)}$ can be upper bounded by a sub-root function*

$$\phi(\rho) = O\left(\sqrt{\frac{\rho N}{n}} \log(nN)\right),$$

*i.e. for any $\rho \gtrsim n^{-2}$*

$$\phi(4\rho) \le 2\phi(\rho), \quad R_n(\mathcal{S}_\rho^{(1)}) \le \phi(\rho).$$

*Proof.* The proof essentially follows from (Lu et al., 2021a, Lemma A.27). First we want to show that $h$ is $C_L$-Lipschitz with respect to $u - u^*$, $\partial_j u - \partial_j u^*$, and $\partial_{jk} u - \partial_{jk} u^*$ for some constant $C_L$. Note that for $h_1 = |\Omega| \cdot (-\mathrm{div}(A\nabla u_1) +$

$Vu_1 - f)^2, h_2 = |\Omega| \cdot (-\mathrm{div}(A\nabla u_2) + Vu_2 - f)^2$ with $u_1, u_2 \in \mathcal{M}_\rho^{(1)}$, we have

$$
\begin{aligned}
|h_1(x) - h_2(x)| &\leq |\Omega| |-\mathrm{div}(A\nabla(u_1 + u_2)) + V(u_1 + u_2) - 2f||-\mathrm{div}(A\nabla(u_1 - u_2)) + V(u_1 - u_2)| \\
&\leq |\Omega| \left| -A \cdot \nabla^2(u_1 + u_2) - \nabla A \cdot \nabla(u_1 + u_2) + V(u_1 + u_2) - 2f \right| \\
&\quad \cdot \left| -A \cdot \nabla^2(u_1 - u_2) - \nabla A \cdot \nabla(u_1 - u_2) + V(u_1 - u_2) \right| \\
&\leq |\Omega| \cdot \left( 2d^2 C_A C + 2d C_A C + 2V_{\max} C \right) \\
&\quad \cdot \left\{ C_A \sum_{j,k=1}^d |(\partial_{jk} u_1(x) - \partial_{jk} u^*(x)) - (\partial_{jk} u_2(x) - \partial_{jk} u^*(x))| \right. \\
&\quad + C_A \sum_{j=1}^d |(\partial_j u_1(x) - \partial_j u^*(x)) - (\partial_j u_2(x) - \partial_j u^*(x))| \\
&\quad \left. + V_{\max} |(u_1(x) - u^*(x)) - (u_2(x) - u^*(x))| \right\}.
\end{aligned}
$$

We can take $C_L = |\Omega| \cdot \left( 2d^2 C_A C + 2d C_A C + 2V_{\max} C \right) \cdot \max\{C_A, V_{\max}\}$.

Denote as the empirical $L^2$-distance $\|u - u^*\|_n = \frac{|\Omega|}{n} \sum_{i=1}^n (u(X_i) - u^*(X_i))^2$. By the uniform boundedness assumption, Lemma C.8 and Dudley's integral theorem, we can estimate the covering number of $\mathcal{S}_\rho^{(1)}$ as follows

$$
\begin{aligned}
&R_n(\mathcal{S}_\rho^{(1)}) \\
=& \mathbb{E}\left[ \sup_{u \in \mathcal{M}_\rho^{(1)}} \frac{1}{n} \sum_{i=1}^n \sigma_i |\Omega| (-\mathrm{div}(A\nabla u)(X_i) + V(X_i)u(X_i) - f(X_i))^2 \right] \\
\leq& \sqrt{2(d^2 + d + 1)} C_L \left\{ \mathbb{E}\left[ \sup_{u \in \mathcal{M}_\rho^{(1)}} \frac{1}{n} \sum_{i=1}^n \sigma_i (u(X_i) - u^*(X_i)) \right] \right. \\
&+ \sum_{j=1}^d \mathbb{E}\left[ \sup_{u \in \mathcal{M}_\rho^{(1)}} \frac{1}{n} \sum_{i=1}^n \sigma_i (\partial_j u(X_i) - \partial_j u^*(X_i)) \right] + \sum_{j,k=1}^d \mathbb{E}\left[ \sup_{u \in \mathcal{M}_\rho^{(1)}} \frac{1}{n} \sum_{i=1}^n \sigma_i (\partial_{jk} u(X_i) - \partial_{jk} u^*(X_i)) \right] \right\} \\
\lesssim& R_n\left( \left\{ (u - u^*) : u \in \mathcal{M}_\rho^{(1)} \right\} \right) + \sum_{j=1}^d R_n\left( \left\{ \partial_j(u - u^*) : u \in \mathcal{M}_\rho^{(1)} \right\} \right) + \sum_{j,k=1}^d R_n\left( \left\{ \partial_{jk}(u - u^*) : u \in \mathcal{M}_\rho^{(1)} \right\} \right) \\
\leq& R_n\left( \left\{ (u - u^*) : u \in \mathcal{F}^{(1)}, \|u - u^*\|_{L^2(\Omega)} \leq \sqrt{\rho} \right\} \right) + \sum_{j=1}^d R_n\left( \left\{ \partial_j(u - u^*) : u \in \mathcal{F}^{(1)}, \|\partial_j(u - u^*)\|_{L^2(\Omega)} \leq \sqrt{\rho} \right\} \right) \\
&+ \sum_{j,k=1}^d R_n\left( \left\{ \partial_{jk}(u - u^*) : u \in \mathcal{F}^{(1)}, \|\partial_{jk}(u - u^*)\|_{L^2(\Omega)} \leq \sqrt{\rho} \right\} \right) \\
\leq& R_n\left( \left\{ (u - u^*) : u \in \mathcal{F}^{(1)}, \|u - u^*\|_n \leq 2\sqrt{\rho} \right\} \right) + \sum_{j=1}^d R_n\left( \left\{ \partial_j(u - u^*) : u \in \mathcal{F}^{(1)}, \|\partial_j(u - u^*)\|_n \leq 2\sqrt{\rho} \right\} \right) \\
&+ \sum_{j,k=1}^d R_n\left( \left\{ \partial_{jk}(u - u^*) : u \in \mathcal{F}^{(1)}, \|\partial_{jk}(u - u^*)\|_n \leq 2\sqrt{\rho} \right\} \right) \\
\lesssim& \inf_{0 < \delta < \sqrt{\rho}} 4\delta + \frac{12}{\sqrt{n_1}} \int_\delta^{\sqrt{\rho}} \sqrt{\log \mathcal{N}(\epsilon, \mathcal{F}^{(1)}, \|\cdot\|_{L^\infty(\Omega)})} d\epsilon + \inf_{0 < \delta < \sqrt{\rho}} 4\delta + \frac{12}{\sqrt{n_1}} \int_\delta^{\sqrt{\rho}} \sqrt{\log \mathcal{N}(\epsilon, \nabla\mathcal{F}^{(1)}, \|\cdot\|_{L^\infty(\Omega), \infty})} d\epsilon \\
&+ \inf_{0 < \delta < \sqrt{\rho}} 4\delta + \frac{12}{\sqrt{n_1}} \int_\delta^{\sqrt{\rho}} \sqrt{\log \mathcal{N}(\epsilon, \nabla^2\mathcal{F}^{(1)}, \|\cdot\|_{L^\infty(\Omega), \infty})} d\epsilon.
\end{aligned}
$$

First note that by the construction $\mathcal{F}^{(1)} = \cup_{W=1}^{W_N} \cup_{S=1}^{S_N} \mathcal{F}^{(1)}(L, W, S, B_N)$, we can upper bound the covering number by

$$
\mathcal{N}(\epsilon, \mathcal{F}^{(1)}, \|\cdot\|_{L^\infty(\Omega)}) \leq W_N S_N \mathcal{N}(\epsilon, \mathcal{F}^{(1)}(L, W_N, S_N, B_N), \|\cdot\|_{L^\infty(\Omega)}).
$$

Also recall (27) that for any $\theta_1, \theta_2 \in \Theta(L, W, S, B)$, it holds that

$$\sup_{x \in \Omega} |\text{clip} \circ f_{\theta_1}(x) - \text{clip} \circ f_{\theta_2}(x)| \vee \sup_{x \in \Omega} \|\nabla(\text{clip} \circ f_{\theta_1})(x) - \nabla(\text{clip} \circ f_{\theta_2})(x)\|_\infty$$

$$\vee \sup_{x \in \Omega} \|\nabla^2(\text{clip} \circ f_{\theta_1})(x) - \nabla^2(\text{clip} \circ f_{\theta_2})(x)\|_\infty$$

$$\leq C_0 W^{\frac{3^L - 1}{2}} (B \vee d)^{\frac{5 \cdot 3^L - 1}{2}} \cdot \|\theta_1 - \theta_2\|_\infty =: C_p \cdot \|\theta_1 - \theta_2\|_\infty$$

To compute the covering number, fix an architecture with $S$ nonzero parameters, and the covering number of this specific architecture with resepect to $\| \cdot \|_{L^\infty(\Omega)}$ can be upper bounded by $\left(\frac{\epsilon}{C_p B}\right)^{-S}$. Also, the number of all possible architectures can be upper bounded by $\binom{2LW^2}{S} \leq (2LW^2)^S$. Hence, we can estimate the covering number

$$\log \mathcal{N}(\epsilon, \mathcal{F}^{(1)}, \| \cdot \|_{L^\infty(\Omega)}) \leq \log \left[ W_N S_N (2LW_N^2)^{S_N} \left(\frac{\epsilon}{C_p B_N}\right)^{-S_N} \right]$$

$$\leq S_N \log(2LW_N^2) + S_N \log(\epsilon^{-1}) + S_N \log\left( C_0 W^{\frac{3^L - 1}{2}} (B \vee d)^{\frac{5 \cdot 3^L - 1}{2}} B_N \right)$$

$$+ \log W_N + \log S_N$$

$$\lesssim N \left[ \log(\epsilon^{-1}) + \log N \right],$$

where the last step follows from that $L = O(1)$, $W_N = O(N)$, $S_N = O(N)$, $B_N = O(N \log N)$. Similarly, we have

$$\log \mathcal{N}(\epsilon, \nabla \mathcal{F}^{(1)}, \| \cdot \|_{L^\infty(\Omega), \infty}) \lesssim N \left[ \log(\epsilon^{-1}) + \log N \right],$$

and

$$\log \mathcal{N}(\epsilon, \nabla^2 \mathcal{F}^{(1)}, \| \cdot \|_{L^\infty(\Omega), \infty}) \lesssim N \left[ \log(\epsilon^{-1}) + \log N \right].$$

Now we can estimate the $R_n(\mathcal{S}_\rho^{(1)})$ by taking $\delta = \frac{1}{n} \lesssim \sqrt{\rho}$ so that

$$R_n(\mathcal{S}_\rho^{(1)}) \lesssim \frac{1}{n} + \frac{1}{\sqrt{n}} \int_{\frac{1}{n}}^{\sqrt{\rho}} \sqrt{N[\log(\epsilon^{-1}) + \log N]} d\epsilon$$

$$\lesssim \frac{1}{n} + \frac{1}{\sqrt{n}} \sqrt{\rho} \sqrt{N[\log n + \log N]}$$

$$\lesssim \sqrt{\frac{\rho N}{n} \log(nN)}.$$

$\square$

**Lemma A.3.** *Consider a Deep neural network space $\mathcal{F}^{(1)}$ on the domain $\Omega$. For any $r > 0$ consider a localized set $\mathcal{M}_r^{(1)}$ defined by*

$$\mathcal{M}_r^{(1)} = \{u \in \mathcal{F}^{(1)} : \|u - u^*\|_{H^2(\Omega)}^2 \leq r\},$$

*and assume the following uniform boundedness condition*

$$\max \left\{ \sup_{u \in \mathcal{F}^{(1)}} \|\nabla^2 u\|_{L^\infty(\Omega), \infty}, \sup_{u \in \mathcal{F}^{(1)}} \|\nabla u\|_{L^\infty(\Omega), \infty}, \sup_{u \in \mathcal{F}^{(1)}} \|u\|_{L^\infty(\Omega)}, \|f\|_{L^\infty(\Omega)} \right\} \leq C.$$

*Furthermore, assume that the Rademacher complexity of a localized set*

$$\mathcal{S}_r^{(1)} = \{h = |\Omega| \cdot (-\text{div}(A\nabla u) + Vu - f)^2 : u \in \mathcal{M}_r^{(1)}\}$$

*can be upper bounded by a sub-root function $\phi(r)$. Let $\{X_i\}_{i=1}^n$ be a sequence of independent uniformly distributed random variables over $\Omega$. Then we have*

$$\sup_{u \in \mathcal{F}^{(1)}} \left[ \| -\text{div}(A\nabla(u - u^*)) + V(u - u^*)\|_{L^2(\Omega)}^2 - \| -\text{div}(A\nabla(u - u^*)) + V(u - u^*)\|_{n,1}^2 \right] \lesssim \max\{r^*, \frac{t}{n}, \frac{t^2}{n^2}\}$$

*with probability at least $1 - e^{-t}$, where $r^*$ is the fixed point of $\phi$.*

*Proof.* The proof essentially follows from (Lu et al., 2021a, Theorem 4.14). Define the following normalized empirical process

$$\tilde{\mathcal{S}}_r^{(1)} = \left\{ \tilde{h} = \frac{\mathbb{E}[h] - h}{\mathbb{E}[h] + r} : h \in \mathcal{S}^{(1)} \right\},$$

and

$$\hat{\mathcal{S}}_r^{(1)} = \left\{ \hat{h} = \frac{h}{\mathbb{E}[h] + r} : h \in \mathcal{S}^{(1)} \right\},$$

where

$$\mathcal{S}^{(1)} = \{ h = |\Omega| \cdot (-\text{div}(A\nabla u) + Vu - f)^2 : u \in \mathcal{F}^{(1)} \}.$$

Then by Lemma C.9,

$$\sup_{\tilde{h} \in \tilde{\mathcal{S}}_r^{(1)}} \mathbb{E}\left[ \frac{1}{n} \sum_{i=1}^n \tilde{h}(X_i) \right] \le \mathbb{E}\left[ \sup_{\tilde{h} \in \tilde{\mathcal{S}}_r^{(1)}} \frac{1}{n} \sum_{i=1}^n \tilde{h}(X_i) \right] \le \mathbb{E}\left[ \sup_{h \in \mathcal{S}^{(1)}} \left| \frac{1}{n} \sum_{i=1}^n \frac{h(X_i) - \mathbb{E}[h]}{\mathbb{E}[h] + r} \right| \right] \le 2R_n(\hat{\mathcal{S}}_r^{(1)}).$$

Moreover, by Lemma C.10

$$R_n(\hat{\mathcal{S}}_r^{(1)}) = \mathbb{E}\left[ \sup_{h \in \mathcal{S}^{(1)}} \frac{1}{n} \sum_{i=1}^n \sigma_i \frac{h(X_i)}{\mathbb{E}[h] + r} \right] \le \frac{4\phi(r)}{r},$$

and consequently

$$\sup_{\tilde{h} \in \tilde{\mathcal{S}}_r^{(1)}} \mathbb{E}\left[ \frac{1}{n} \sum_{i=1}^n \tilde{h}(X_i) \right] \le \frac{8\phi(r)}{r}.$$

Next we want to apply Lemma C.7 to $\tilde{h} \in \tilde{S}_r^{(1)}$. First we check that $\tilde{h}$ is uniformly bounded. Indeed

$$\|\tilde{h}\|_\infty = \frac{\|\mathbb{E}[h] - h\|_\infty}{|\mathbb{E}[h] + r|} \le \frac{2\|h\|_\infty}{r} = \frac{2|\Omega| \cdot \| - \text{div}(A\nabla u) + Vu - f\|_\infty^2}{r} \le \frac{C_1}{r},$$

where we can take $C_1 = 2|\Omega|C^2(d^2 C_A + d C_A + 1)^2$. Next from definition it is evident that

$$\mathbb{E}[\tilde{h}] = \mathbb{E}\left[ \frac{\mathbb{E}[h] - h}{\mathbb{E}[h] + r} \right] = 0.$$

Finally, we need to check that the second moment of $\tilde{h}$ is uniformly bounded. To see this,

$$\mathbb{E}[\tilde{h}^2] = \mathbb{E}\left[ \frac{|\mathbb{E}[h] - h|^2}{|\mathbb{E}[h] + r|^2} \right] \le \frac{\|h\|_\infty \mathbb{E}[h]}{2\mathbb{E}[h]r} \le \frac{C_1}{r} =: \sigma^2,$$

Let $U = \max\{\sigma, \frac{2C|\Omega|}{r}\} = \max\{\sqrt{\frac{C_1}{r}}, \frac{C_1}{r}\} \ge \sigma$. Hence the conditions of Lemma C.7 are satisfied. Take $\{X_i\}_{i=1}^n$ as independent uniform random variables on $\Omega$, and we have with probability at least $1 - e^{-t}$,

$$\sup_{\tilde{h} \in \tilde{S}_r^{(1)}} \frac{1}{n} \sum_{i=1}^n \tilde{h}(X_i) \le 2\mathbb{E}\left[ \sup_{\tilde{h} \in \tilde{S}_r^{(1)}} \frac{1}{n} \sum_{i=1}^n \tilde{h}(X_i) \right] + \frac{4Ut}{3n} + \sqrt{\frac{2\sigma^2 t}{n}}$$

$$\le \frac{16\phi(r)}{r} + \max\left\{ \sqrt{\frac{16C_1 t^2}{9rn^2}}, \frac{4C_1 t}{3rn} \right\} + \sqrt{\frac{2C_1 t}{rn}}.$$

Take $r_0 = \max\{2^{14} r^*, \frac{1024 C_1 t^2}{9n^2}, \frac{128 C_1 t}{n}\}$, where $r^*$ is the fixed point of $\phi$. Since $\phi$ is concave, we have $\phi(r) \le r$ if $r \ge r^*$. Together with the fact that $\phi(4r) \le 2\phi(r)$, we have

$$\frac{16\phi(r_0)}{r_0} \le \frac{2^{11}\phi(\frac{r_0}{2^{14}})}{2^{14}\frac{r_0}{2^{14}}} \le \frac{1}{8}.$$

Also, by the definition of $r_0$

$$\sqrt{\frac{16C_1t^2}{9r_0n^2}} \vee \frac{4C_1t}{3r_0n} \vee \sqrt{\frac{2C_1t}{r_0n}} \leq \frac{1}{8},$$

Consequently, we have for any $\tilde{h} \in \tilde{S}_r^{(1)}$

$$\frac{1}{n}\sum_{i=1}^n \frac{\mathbb{E}[h] - h(X_i)}{\mathbb{E}[h] + r_0} = \frac{1}{n}\sum_{i=1}^n \tilde{h}(X_i) \leq \frac{1}{8} + \frac{1}{8} + \frac{1}{8} \leq \frac{1}{2},$$

or equivalently, for any $h \in S^{(1)}$,

$$\mathbb{E}[h] \leq \frac{2}{n}\sum_{i=1}^n h(X_i) + r_0,$$

$$\mathbb{E}[h] \lesssim \frac{1}{n}\sum_{i=1}^n h(X_i) + \max\{r^*, \frac{t}{n}, \frac{t^2}{n^2}\},$$

with probability at least $1 - e^{-t}$. □

**Lemma A.4.** *Consider a Deep neural network space $\mathcal{F}^{(2)} = \cup_{W=1}^{W_N} \cup_{S=1}^{S_N} \mathcal{F}(L, W, S, B_N)$ on $\partial\Omega$ with $L = O(1)$, $W_N = O(N)$, $S_N = O(N)$, $B_N = O(N \log N)$, where $N \in \mathbb{N}$ is fixed to be sufficiently large. For any $\rho > 0$, consider a localized set $\mathcal{M}_\rho^{(2)}$ defined by*

$$\mathcal{M}_\rho^{(2)} = \{u \in \mathcal{F}^{(2)} : \|u - u^*\|_{L^\infty(\partial\Omega)}^2 \leq \rho\},$$

*and a localized set $S_\rho^{(2)}$ defined by*

$$S_\rho^{(2)} = \{h = |\partial\Omega| \cdot (u - u^*)^2 : u \in \mathcal{M}_\rho^{(2)}\}.$$

*Further assume the following uniform boundedness condition*

$$\max\{\sup_{u \in \mathcal{F}^{(2)}} \|u\|_{L^\infty(\partial\Omega)}, \|u^*\|_{L^\infty(\partial\Omega)}\} \leq C.$$

*Then for any $\rho \gtrsim n^{-2}$, the Rademacher complexity of $S_\rho^{(2)}$ can be upper bounded by a sub-root function*

$$\phi(\rho) = O\left(\sqrt{\frac{\rho N}{n} \log(nN)}\right),$$

*i.e. for any $\rho \gtrsim n^{-2}$*

$$\phi(4\rho) \leq 2\phi(\rho), \quad R_n(S_\rho^{(2)}) \leq \phi(\rho).$$

*Proof.* The proof is similar to that of Lemma A.2. □

**Lemma A.5.** *Consider a Deep neural network space $\mathcal{F}^{(2)}$ on $\partial\Omega$. For any $r > 0$ consider a localized set $\mathcal{M}_r^{(2)}$ defined by*

$$\mathcal{M}_r^{(2)} = \{u \in \mathcal{F}^{(2)} : \|u - u^*\|_{L^\infty(\partial\Omega)}^2 \leq r\},$$

*and assume the following uniform boundedness condition*

$$\max\{\sup_{u \in \mathcal{F}^{(2)}} \|u\|_{L^\infty(\partial\Omega)}, \|u^*\|_{L^\infty(\partial\Omega)}\} \leq C.$$

*Furthermore, assume that the Rademacher complexity of a localized set*

$$S_r^{(2)} = \{h = |\partial\Omega| \cdot (u - u^*)^2 : u \in \mathcal{M}_r^{(2)}\}$$

*can be upper bounded by a sub-root function $\phi(r)$. Let $\{Y_j\}_{j=1}^n$ be a sequence of independent uniformly distributed random variable over $\partial\Omega$. Then we have*

$$\sup_{u \in \mathcal{F}^{(2)}} \left[ \|u - u^*\|_{L^2(\partial\Omega)}^2 - \|u - u^*\|_{n,2}^2 \right] \lesssim \max\{r^*, \frac{t}{n}, \frac{t^2}{n^2}\}$$

*with probability at least $1 - e^{-t}$, where $r^*$ is the fixed point of $\phi$.*

*Proof.* The proof is similar to that of Lemma A.3. $\square$

*Proof of Proposition 4.3.* Take $\mathcal{F}^{(1)}$ in Lemma A.2 to be the restriction of the sieve $\mathcal{F}_n$ on $\Omega$. Note that $\mathcal{F}^{(1)}$ fulfills the uniform boundedness assumption due to the presence of the cutoff function. By the construction of the sieve, we have $N \asymp n^{\frac{d}{d+2(\beta-2)}}$ so that

$$R_n(\mathcal{S}_r^{(1)}) \lesssim \sqrt{rn^{\frac{-2(\beta-2)}{d+2(\beta-2)}} \log n} = \phi(r),$$

and the fixed $r^*$ of $\phi$ satisfies

$$r^* \asymp n^{\frac{-2(\beta-2)}{d+2(\beta-2)}} \log n = \epsilon_n^2.$$

By taking $t = n^{\frac{d}{d+2(\beta-2)}} \log n = n\epsilon_n^2$, Lemma A.3 shows that

$$\sup_{u \in \mathcal{F}_n} \left[ \| - \operatorname{div}(A\nabla(u - u^*)) + V(u - u^*)\|_{L^2(\Omega)}^2 - \| - \operatorname{div}(A\nabla(u - u^*)) + V(u - u^*)\|_{n,1}^2 \right] \lesssim \max\{r^*, \frac{t}{n}, \frac{t^2}{n^2}\} \lesssim \epsilon_n^2,$$
(30)

with probability at least $1 - e^{-n\epsilon_n^2}$.

Take $\mathcal{F}^{(2)}$ in Lemma A.4 to be the restriction of $\mathcal{F}_n$ on $\partial\Omega$. Analogously, Lemma A.5 yields

$$\sup_{u \in \mathcal{F}_n} \left[ \|u - u^*\|_{L^2(\partial\Omega)}^2 - \|u - u^*\|_{n,2}^2 \right] \lesssim \epsilon_n^2,$$
(31)

with probability at least $1 - e^{-n\epsilon_n^2}$. We can conclude the proof by combining (30) and (31). $\square$

# B. Proof of the main results

### B.1. Proof of Theorem 3.1

*Proof of Theorem 3.1.* The proof follows by combining Lemma 4.1 and Proposition 4.3. Recall that in the proof of Lemma 4.1, we have (19)

$$P_{u^*}^{(n)} Q_{u^*}^{(n)} \left[ \Pi(u \in \mathcal{F}_n : \| - \operatorname{div}(A\nabla(u - u^*)) + V(u - u^*)\|_{n,1} > JM\epsilon_n | \mathcal{D}^{(n)}) \right]$$

$$\leq e^{Cn\epsilon_n^2} \frac{e^{-\frac{1}{32}nM^2\epsilon_n^2}}{1 - e^{-\frac{1}{32}nM^2\epsilon_n^2}} + e^{-\frac{1}{36}nM^2\epsilon_n^2 J^2 + \xi n\epsilon_n^2 + (1+C_1)n\epsilon_n^2} + C_1^{-1}(n\epsilon_n^2)^{-1}.$$

Also, take $J, M$ to be large enough so that (30) implies that for any $u \in \mathcal{F}_n$,

$$\| - \operatorname{div}(A\nabla(u - u^*)) + V(u - u^*)\|_{L^2(\Omega)}^2 - \| - \operatorname{div}(A\nabla(u - u^*)) + V(u - u^*)\|_{n,1}^2 \leq J^2 M^2 \epsilon_n^2,$$

with probability at least $1 - e^{-n\epsilon_n^2}$. Combining the two inequalities above yields

$$P_{u^*}^{(n)} Q_{u^*}^{(n)} \left[ \Pi(u \in \mathcal{F}_n : \| - \operatorname{div}(A\nabla(u - u^*)) + V(u - u^*)\|_{L^2(\Omega)} > 2JM\epsilon_n | \mathcal{D}^{(n)}) \right]$$

$$\leq P_{u^*}^{(n)} Q_{u^*}^{(n)} \left[ \Pi(u \in \mathcal{F}_n : \| - \operatorname{div}(A\nabla(u - u^*)) + V(u - u^*)\|_{n,1} > JM\epsilon_n | \mathcal{D}^{(n)}) \right] + e^{-n\epsilon_n^2}$$
(32)

$$\leq e^{Cn\epsilon_n^2} \frac{e^{-\frac{1}{32}nM^2\epsilon_n^2}}{1 - e^{-\frac{1}{32}nM^2\epsilon_n^2}} + e^{-\frac{1}{36}nM^2\epsilon_n^2 J^2 + \xi n\epsilon_n^2 + (1+C_1)n\epsilon_n^2} + C_1^{-1}(n\epsilon_n^2)^{-1} + e^{-n\epsilon_n^2}.$$

Similarly, by leveraging (20) and (31) we have

$$P_{u^*}Q_{u^*}\left[\Pi(u \in \mathcal{F}_n : \|u - u^*\|_{L^2(\partial\Omega)} > 2JM\epsilon_n|\mathcal{D}^{(n)})\right]$$

$$\leq e^{Cn\epsilon_n^2}\frac{e^{-\frac{1}{32}nM^2\epsilon_n^2}}{1 - e^{-\frac{1}{32}nM^2\epsilon_n^2}} + e^{-\frac{1}{36}nM^2\epsilon_n^2 J^2 + \xi n\epsilon_n^2 + (1+C_1)n\epsilon_n^2} + C_1^{-1}(n\epsilon_n^2)^{-1} + e^{-n\epsilon_n^2}. \tag{33}$$

We can then conclude by putting (32), (33) and (22) together. □

## B.2. Proof of Corollary 3.2

*Proof of Corollary 3.2.* Note that it suffices to show that for any $M_n \to \infty$,

$$P_{u^*}^{(n)}Q_{u^*}^{(n)}\left(u \in \mathcal{F} : \|-\operatorname{div}(A\nabla\bar{u}_n) + V\bar{u}_n - f\|_{L^2(\Omega)}^2 + \lambda\|\bar{u}_n - g\|_{L^2(\partial\Omega)}^2 \geq M_n^2\epsilon_n^2\right) \to 0,$$

as $n \to \infty$. Also by Jensen's inequality, we have

$$\|-\operatorname{div}(A\nabla\bar{u}_n) + V\bar{u}_n - f\|_{L^2(\Omega)} \leq \mathbb{E}_{u\sim\Pi}\left[\|-\operatorname{div}(A\nabla u) + Vu - f\|_{L^2(\Omega)}|\mathcal{D}^{(n)}\right],$$

and

$$\|\bar{u}_n - g\|_{L^2(\partial\Omega)} \leq \mathbb{E}_{u\sim\Pi}\left[\|u - g\|_{L^2(\partial\Omega)}|\mathcal{D}^{(n)}\right].$$

Hence to prove the statement, it is sufficient to show that for any $M_n \to \infty$

$$P_{u^*}^{(n)}Q_{u^*}^{(n)}\left(\mathbb{E}_{u\sim\Pi}\left[\|-\operatorname{div}(A\nabla u) + Vu - f\|_{L^2(\Omega)}|\mathcal{D}^{(n)}\right] > M_n\epsilon_n\right) \to 0, \tag{34}$$

and

$$P_{u^*}^{(n)}Q_{u^*}^{(n)}\left(\mathbb{E}_{u\sim\Pi}\left[\|u - g\|_{L^2(\partial\Omega)}|\mathcal{D}^{(n)}\right] > M_n\epsilon_n\right) \to 0, \tag{35}$$

as $n \to \infty$.

By Cauchy-Schwarz and Jensen's inequality,

$$\mathbb{E}_{u\sim\Pi}\left[\|-\operatorname{div}(A\nabla u) + Vu - f\|_{L^2(\Omega)}|\mathcal{D}^{(n)}\right]$$

$$\leq 2JM\epsilon_n + \mathbb{E}_{u\sim\Pi}\left[\|-\operatorname{div}(A\nabla u) + Vu - f\|_{L^2(\Omega)}\mathbf{1}_{\{\|-\operatorname{div}(A\nabla u)+Vu-f\|_{L^2(\Omega)}>2JM\epsilon_n\}}|\mathcal{D}^{(n)}\right]$$

$$\leq 2JM\epsilon_n + \left(\mathbb{E}\left[\|-\operatorname{div}(A\nabla u) + Vu - f\|_{L^2(\Omega)}^2|\mathcal{D}^{(n)}\right]\right)^{1/2}\left(\Pi\left[\|-\operatorname{div}(A\nabla u) + Vu - f\|_{L^2(\Omega)} > 2JM\epsilon_n|\mathcal{D}^{(n)}\right]\right)^{1/2}.$$
$$\tag{36}$$

Let $\phi_1$ be test on $\{X_i, f_i\}_{i=1}^n$ such that (14) holds, and let $E$ to be the set on which (17) holds. Recall that by (15), (16) and (17)

$$P_{u^*}^{(n)}Q_{u^*}^{(n)}\left[\Pi(u \in \mathcal{F}_n : \|-\operatorname{div}(A\nabla(u-u^*)) + V(u-u^*)\|_{n,1} > JM\epsilon_n|\mathcal{D}^{(n)})\mathbf{1}_E\right]$$

$$\leq P_{u^*}^{(n)}Q_{u^*}^{(n)}(\phi_1) + P_{u^*}^{(n)}Q_{u^*}^{(n)}\left[\Pi(u \in \mathcal{F}_n : \|-\operatorname{div}(A\nabla(u-u^*)) + V(u-u^*)\|_{n,1} > JM\epsilon_n|\mathcal{D}^{(n)})(1-\phi_1)\mathbf{1}_E\right]$$

$$\leq e^{Cn\epsilon_n^2}\frac{e^{-\frac{1}{32}nM^2\epsilon_n^2}}{1 - e^{-\frac{1}{32}nM^2\epsilon_n^2}} + e^{-\frac{1}{36}nM^2\epsilon_n^2 J^2 + \xi n\epsilon_n^2 + (1+C_1)n\epsilon_n^2}.$$

Then by (32), we can take $J, M$ to be large enough such that

$$P_{u^*}^{(n)}Q_{u^*}^{(n)}\left[\Pi(u \in \mathcal{F}_n : \|-\operatorname{div}(A\nabla(u-u^*)) + V(u-u^*)\|_{L^2(\Omega)} > 2JM\epsilon_n|\mathcal{D}^{(n)})\mathbf{1}_E\right]$$

$$\leq P_{u^*}^{(n)}Q_{u^*}^{(n)}\left[\Pi(u \in \mathcal{F}_n : \|-\operatorname{div}(A\nabla(u-u^*)) + V(u-u^*)\|_{n,1} > JM\epsilon_n|\mathcal{D}^{(n)})\mathbf{1}_E\right] + e^{-n\epsilon_n^2 J^2}$$

$$\leq e^{Cn\epsilon_n^2}\frac{e^{-\frac{1}{32}nM^2\epsilon_n^2}}{1 - e^{-\frac{1}{32}nM^2\epsilon_n^2}} + e^{-\frac{1}{36}nM^2\epsilon_n^2 J^2 + \xi n\epsilon_n^2 + (1+C_1)n\epsilon_n^2} + e^{-n\epsilon_n^2 J^2}. \tag{37}$$

Furthermore, by Markov's inequality

$$P_{u^*}^{(n)} Q_{u^*}^{(n)} \left( \Pi(u \in \mathcal{F}_n : \| - \mathrm{div}(A\nabla u) + Vu - f\|_{L^2(\Omega)} > 2JM\epsilon_n | \mathcal{D}^{(n)}) \mathbf{1}_E > e^{-bn\epsilon_n^2} \right)$$
$$\leq e^{bn\epsilon_n^2} \left( e^{Cn\epsilon_n^2} \frac{e^{-\frac{1}{32}nM^2\epsilon_n^2}}{1 - e^{-\frac{1}{32}nM^2\epsilon_n^2}} + e^{-\frac{1}{36}nM^2\epsilon_n^2 J^2 + \xi n\epsilon_n^2 + (1+C_1)n\epsilon_n^2} + e^{-n\epsilon_n^2 J^2} \right). \tag{38}$$

Now we can estimate the second term in (36) by

$$P_{u^*}^{(n)} Q_{u^*}^{(n)} \left( \mathbb{E}\left[ \| - \mathrm{div}(A\nabla u) + Vu - f\|_{L^2(\Omega)}^2 | \mathcal{D}^{(n)} \right] \cdot \Pi\left[ \| - \mathrm{div}(A\nabla u) + Vu - f\|_{L^2(\Omega)} > 2JM\epsilon_n | \mathcal{D}^{(n)} \right] > \epsilon_n^2 \right)$$
$$\leq P_{u^*}^{(n)} Q_{u^*}^{(n)} \left( \mathbb{E}\left[ \| - \mathrm{div}(A\nabla u) + Vu - f\|_{L^2(\Omega)}^2 | \mathcal{D}^{(n)} \right] \cdot \Pi\left[ \| - \mathrm{div}(A\nabla u) + Vu - f\|_{L^2(\Omega)} > 2JM\epsilon_n | \mathcal{D}^{(n)} \right] \mathbf{1}_E > \epsilon_n^2 \right)$$
$$\quad + P_{u^*}^{(n)} Q_{u^*}^{(n)}(\mathbf{1}_{E^c})$$
$$\leq P_{u^*}^{(n)} Q_{u^*}^{(n)} \left( \mathbb{E}\left[ \| - \mathrm{div}(A\nabla u) + Vu - f\|_{L^2(\Omega)}^2 | \mathcal{D}^{(n)} \right] \mathbf{1}_E \cdot e^{-bn\epsilon_n^2} > \epsilon_n^2 \right)$$
$$\quad + e^{bn\epsilon_n^2} \left( e^{Cn\epsilon_n^2} \frac{e^{-\frac{1}{32}nM^2\epsilon_n^2}}{1 - e^{-\frac{1}{32}nM^2\epsilon_n^2}} + e^{-\frac{1}{36}nM^2\epsilon_n^2 J^2 + \xi n\epsilon_n^2 + (1+C_1)n\epsilon_n^2} + e^{-n\epsilon_n^2 J^2} \right) + C_1^{-1}(n\epsilon_n^2)^{-1}$$
$$= P_{u^*}^{(n)} Q_{u^*}^{(n)} \left( \frac{\int \| - \mathrm{div}(A\nabla u) + Vu - f\|_{L^2(\Omega)}^2 \frac{p_u^{(n)}}{p_{u^*}^{(n)}} \frac{q_u^{(n)}}{q_{u^*}^{(n)}} d\Pi(u)}{\int \frac{p_u^{(n)}}{p_{u^*}^{(n)}} \frac{q_u^{(n)}}{q_{u^*}^{(n)}} d\Pi(u)} \mathbf{1}_E > e^{bn\epsilon_n^2} \epsilon_n^2 \right)$$
$$\quad + e^{bn\epsilon_n^2} \left( e^{Cn\epsilon_n^2} \frac{e^{-\frac{1}{32}nM^2\epsilon_n^2}}{1 - e^{-\frac{1}{32}nM^2\epsilon_n^2}} + e^{-\frac{1}{36}nM^2\epsilon_n^2 J^2 + \xi n\epsilon_n^2 + (1+C_1)n\epsilon_n^2} + e^{-n\epsilon_n^2 J^2} \right) + C_1^{-1}(n\epsilon_n^2)^{-1}$$
$$\leq e^{\xi n\epsilon_n^2 + (1+C_1)n\epsilon_n^2} e^{-bn\epsilon_n^2} \epsilon_n^{-2} \int \| - \mathrm{div}(A\nabla u) + Vu - f\|_{L^2(\Omega)}^2 d\Pi(u)$$
$$\quad + e^{bn\epsilon_n^2} \left( e^{Cn\epsilon_n^2} \frac{e^{-\frac{1}{32}nM^2\epsilon_n^2}}{1 - e^{-\frac{1}{32}nM^2\epsilon_n^2}} + e^{-\frac{1}{36}nM^2\epsilon_n^2 J^2 + \xi n\epsilon_n^2 + (1+C_1)n\epsilon_n^2} + e^{-n\epsilon_n^2 J^2} \right) + C_1^{-1}(n\epsilon_n^2)^{-1}. \tag{39}$$

where the second inequality follows from (38), the third inequality follows from Markov's inequality, Fubini's theorem and (17). Then we can take $C_1 = 1$, $b = \xi + 1 + C_1$ and $M, J$ to be some large constant such that (39) goes to 0 as $n \to \infty$. Together with (36), we prove (34). (35) can argued in a similar way. Therefore we conclude the proof. $\square$

### B.3. Proof of Theorem 3.5

*Proof of Theorem 3.5.* Note that by the definition of the PINN-loss $\mathcal{E}(\cdot)$, we have

$$2\inf_\psi \sup_{u^*} \mathcal{E}(\psi(\mathcal{D})) \geq \inf_\psi \sup_{u^*} \mathbb{E}\| - \mathrm{div}(A\nabla\psi(\mathcal{D})) + V\psi(\mathcal{D}) - f\|_{L^2(\Omega)}^2 + \lambda \inf_\psi \sup_{u^*} \mathbb{E}\|\psi(\mathcal{D}) - g\|_{L^2(\partial\Omega)}^2.$$

The statement follows directly from Lemma B.1 and Lemma B.2. $\square$

**Lemma B.1.** *Assume the same setting as in Theorem 3.5. Then we have*

$$\inf_\psi \sup_{u^*} \mathbb{E}\| - \mathrm{div}(A\nabla\psi(\mathcal{D})) + V\psi(\mathcal{D}) - f\|_{L^2(\Omega)}^2 \gtrsim n_1^{-\frac{2(\beta-2)}{d+2(\beta-2)}},$$

*where the supremum is taken over all $u^* \in C^\beta(\bar{\Omega})$ which is a solution to (1) with $f \in C^{\beta-2}(\bar{\Omega})$ $g \in C^\beta(\partial\Omega)$ such that $\|u^*\|_{C^\beta(\bar{\Omega})} \leq K$, and the infimum is taken over all estimators $\psi : (\mathbb{R}^d)^{\otimes n_1} \times \mathbb{R}^{\otimes n_1} \times (\mathbb{R}^d)^{\otimes n_2} \times \mathbb{R}^{\otimes n_2} \to C^\beta(\bar{\Omega})$.*

*Proof.* Since $\Omega$ is open and nonempty, there exists a cube $B(x_0, r) = \{x \in \Omega : \|x - x_0\|_\infty < r\}$ with center $x_0 \in \Omega$ and radius $r > 0$ such that $B(x_0, r) \subset\subset \Omega$. Consider a bump function $g \in C^\infty(\mathbb{R}^d)$ with $\mathrm{supp}(g) \subset B(x_0, r)$ and $\|g\|_{C^\beta(\bar{\Omega})} \leq K$. Moreover, consider a uniform partition over $B(x_0, r)$ with grids $x^{(j)}, j \in [m]^d$, where the partition size $m$ is

to be determined later. By Varshamov-Gilbert lemma (Tsybakov, 2009), there exist a sequence $\tau^{(1)}, \ldots, \tau^{(2^{m^d/8})} \in \{0,1\}^d$ such that $\|\tau^{(k)} - \tau^{(k')}\|_1 \geq \frac{m^d}{8}$ for all $k, k' \in \{0,1\}^d, k \neq k'$. We construct a packing as

$$u_k(x) = \sum_{j \in [m]^d} \tau_j^{(k)} \frac{w}{m^\beta} g(m(x - x^{(j)})), \quad k \in [2^{m^d/8}],$$

where the constant $w < 1$ is to be determined. One can easily check that $\|u_k\|_{C^\beta(\bar{\Omega})} \leq K$ for all $k \in [2^{m^d/8}]$.

Furthermore, the distance of two distinct functions in the packing can be lower bounded as

$$\| - \operatorname{div}(A\nabla(u_k - u_k')) + V(u_k - u_k')\|_{L^2(\Omega)}^2 \gtrsim \frac{w^2}{m^{2\beta-4+d}} \|\tau^{(k)} - \tau^{(k')}\|_1 \|\nabla^2 g\|_{L^2(\mathbb{R}^d)}^2 \geq \frac{w^2}{8m^{2\beta-4}} \|\nabla^2 g\|_{L^2(\mathbb{R}^d)}^2 =: (2\delta)^2.$$

By Fano's method, we have

$$\inf_\psi \sup_{u^*} \mathbb{E}\| - \operatorname{div}(A\nabla\psi(\mathcal{D})) + V\psi(\mathcal{D}) - f\|_{L^2(\Omega)}^2 \geq \delta^2 \left(1 - \frac{I(V; \mathcal{D}) + \log 2}{\log(2^{m^d/8})}\right),$$

where $V \sim \mathrm{U}(\{1, 2, \ldots, 2^{m^d/8}\})$, and we can upper bound the mutual information $I(V; \mathcal{D})$ using local Fano's method

$$I(V; \mathcal{D}) \leq \frac{1}{|2^{m^d/8}|^2} \sum_{k \neq k'} D_{\mathrm{KL}}(P_k \| P_{k'}) \leq \max_{k \neq k'} D_{\mathrm{KL}}(P_k, P_{k'}),$$

where $P_k$ is the joint distribution of $\mathcal{D} = \{X_i, f_i\}_{i=1}^{n_1} \cup \{Y_j, g_j\}_{j=1}^{n_2}$ conditioned on $V = k$. Then we can compute

$$D_{\mathrm{KL}}(P_k \| P_{k'}) = \frac{n_1}{2|\Omega|} \| - \operatorname{div}(A\nabla(u_k - u_k')) + V(u_k - u_k')\|_{L^2(\Omega)}^2 + \frac{n_2}{2|\partial\Omega|} \|u_k - u_{k'}\|_{L^2(\partial\Omega)}^2$$

$$\leq Cn_1 \frac{w^2}{m^{2\beta-4}},$$

where the second term vanishes since $u_k$'s are supported on $B(x_0, r) \subset\subset \Omega$. Hence choose $m = \lceil n_1^{\frac{1}{d+2(\beta-2)}} \rceil$ and $w$ to be sufficiently small so that

$$\frac{I(V; \mathcal{D}) + \log 2}{\log(2^{m^d/8})} \leq 16C \frac{n_1 w^2}{m^{2\beta+d-4} \log 2} \leq \frac{1}{2}.$$

Hence we obtain a lower bound

$$\inf_\psi \sup_{u^*} \mathbb{E}\| - \operatorname{div}(A\nabla\psi(\mathcal{D})) + V\psi(\mathcal{D}) - f\|_{L^2(\Omega)}^2 \gtrsim \frac{1}{m^{2\beta-4}} \asymp n_1^{-\frac{2(\beta-2)}{d+2(\beta-2)}}.$$

$\square$

**Lemma B.2.** *Assume the same setting as in Theorem 3.5. Then we have*

$$\inf_\psi \sup_{u^*} \mathbb{E}\|\psi(\mathcal{D}) - g\|_{L^2(\partial\Omega)}^2 \gtrsim n_2^{-\frac{2\beta}{d-1+2\beta}},$$

*where the supremum is taken over all $u^* \in C^\beta(\bar{\Omega})$ which is a solution to (1) with $f \in C^{\beta-2}(\bar{\Omega})$ $g \in C^\beta(\partial\Omega)$ such that $\|u^*\|_{C^\beta(\bar{\Omega})} \leq K$, and the infimum is taken over all estimators $\psi : (\mathbb{R}^d)^{\otimes n_1} \times \mathbb{R}^{\otimes n_1} \times (\mathbb{R}^d)^{\otimes n_2} \times \mathbb{R}^{\otimes n_2} \to C^\beta(\bar{\Omega})$.*

*Proof.* We first straighten the boundary. Take $x_0 \in \partial\Omega$ and let $\phi : B \to \partial\Omega \cap U$ be a $C^\infty$-bijection such that both $\phi$ and $\phi^{-1}$ is $C^\infty$, where $U$ is an neighborhood of $x_0$ and $B \subset \mathbb{R}^{d-1}$ is a open cube. Consider a bump function $g \in C^\infty(\mathbb{R}^{d-1})$ with $\operatorname{supp}(g) \subset B$. Consider a uniform partition over $B$ with grids $y^{(j)}, j \in [m]^{d-1}$, where the partition size $m$ is to be determined. By Varshamov-Gilbert lemma (Tsybakov, 2009), there exists a sequence $\tau^{(1)}, \ldots, \tau^{(2^{m^{d-1}/8})} \in \{0,1\}^{d-1}$ such that $\|\tau^{(k)} - \tau^{(k')}\|_1 \geq \frac{m^{d-1}}{8}$ for all $k, k' \in [m]^{d-1}, k \neq k'$. We construct a packing

$$g_k(y) = \sum_{j \in [m]^{d-1}} \tau_j^{(k)} \frac{w}{m^\beta} g(m(y - y^{(j)})), \quad k \in [2^{m^{d-1}/8}],$$

and for each $k \in [2^{m^{d-1}/8}]$, define

$$\bar{g}_k(x) := \begin{cases} g_k \circ \phi^{-1}(x), & x \in \partial\Omega \cap U, \\ 0 & x \in \partial\Omega \backslash U, \end{cases}$$

and $\bar{g}_k \in C^\beta(\partial\Omega)$ since $\phi^{-1}$ is $C^\infty$. Moreover, let $u_k \in C^\beta(\bar{\Omega})$ be the solution to the PDE

$$-\text{div}(A\nabla u_k) + Vu_k = 0 \quad \text{in } \Omega,$$
$$u_k = \bar{g}_k \quad \text{on } \partial\Omega,$$

and by choosing $w$ to be sufficiently small, it is easy to guarantee that $\|u_k\|_{C^\beta(\bar{\Omega})} \le K$ for all $k \in [2^{m^{d-1}/8}]$.

The distance of two distinct functions in the packing can be lower bounded as

$$\|u_k - u_{k'}\|^2_{L^2(\partial\Omega)} = \|\bar{g}_k - \bar{g}_{k'}\|^2_{L^2(\partial\Omega)} \gtrsim \|g_k - g_{k'}\|^2_{L^2(B)} \gtrsim \frac{w^2}{m^{2\beta+d-1}}\|\tau^{(k)} - \tau^{(k')}\|_1 \|g\|^2_{L^2(B)}$$

$$\ge \frac{w^2}{8m^{2\beta}}\|g\|^2_{L^2(B)} =: (2\delta)^2.$$

By Fano's method, we have

$$\inf_\psi \sup_{u^*} \mathbb{E}\|\psi(\mathcal{D}) - g\|^2_{L^2(\partial\Omega)} \ge \delta^2 \left(1 - \frac{I(V;\mathcal{D}) + \log 2}{\log\left(2^{m^{d-1}/8}\right)}\right),$$

where $V \sim U(\{1, 2, \ldots, 2^{m^{d-1}/8}\})$, and we can upper bound the mutual information $I(V;\mathcal{D})$ using local Fano's method

$$I(V;\mathcal{D}) \le \frac{1}{|2^{m^{d-1}/8}|^2} \sum_{k \ne k'} D_{\text{KL}}(P_k||P_{k'}) \le \max_{k,k'} D_{\text{KL}}(P_k, P_{k'}),$$

where $P_k$ is the joint distribution of $\mathcal{D} = \{X_i, f_i\}^{n_1}_{i=1} \cup \{Y_j, g_j\}^{n_2}_{j=1}$ conditioned on $V = k$. Then we can compute

$$D_{\text{KL}}(P_k||P_{k'}) = \frac{n_1}{2|\Omega|}\| - \text{div}(A\nabla(u_k - u_k')) + V(u_k - u_k')\|^2_{L^2(\Omega)} + \frac{n_2}{2|\partial\Omega|}\|u_k - u_{k'}\|^2_{L^2(\partial\Omega)}$$

$$\le Cn_2 \frac{w^2}{m^{2\beta}},$$

where the first term vanishes since $\Delta u_k = 0$ in $\Omega$. Then we can choose $m = \lceil n_2^{\frac{1}{d-1+2\beta}} \rceil$ and $w$ to be sufficiently small so that

$$\frac{I(V;\mathcal{D}) + \log 2}{\log\left(2^{m^{d-1}/8}\right)} \le 16C\frac{n_2w^2}{m^{2\beta+d-1}\log 2} \le \frac{1}{2},$$

and we obtain a lower bound

$$\inf_\psi \sup_{u^*} \mathbb{E}\|\psi(\mathcal{D}) - g\|^2_{L^2(\partial\Omega)} \gtrsim \frac{1}{m^{2\beta}} \asymp n_2^{-\frac{2\beta}{d-1+2\beta}}.$$

$\square$

## C. Auxiliary tools

### C.1. Auxiliary tools for Lemma 4.1

**Hypothesis testing condition** Let $\Theta$ be a parameter space and $\theta_0 \in \Theta$ be the ground truth. Let $d_n$ and $e_n$ be semimetrics on $\Theta$ such that there exists $K > 0$ and $\xi > 0$ such that for any $\epsilon > 0$ and for any $\theta_1 \in \Theta$ with $d_n(\theta_1, \theta_0) > \epsilon$, there exists a test $\phi_n$ such that

$$P^{(n)}_{\theta_0}\phi_n \le e^{-Kn\epsilon^2}, \qquad \sup_{\theta:e_n(\theta,\theta_1)<\xi\epsilon} P^{(n)}_\theta(1 - \phi_n) \le e^{-Kn\epsilon^2}. \tag{40}$$

The following result shows that the Euclidean distance satisfies the hypothesis testing condition with $\xi = \frac{1}{2}$ and $K = \frac{1}{32}$ under the gaussian regression setting.

**Lemma C.1** ((Ghosal & Van der Vaart, 2017, Lemma 8.27)). *For $\theta \in \mathbb{R}^n$ and $\sigma > 0$, let $P_\theta^{(n)} = \mathcal{N}(\theta, \sigma^2 I_n)$. Then for any $\theta_0, \theta_1 \in \mathbb{R}^n$, the test $\phi_n = \mathbf{1}_{\{\theta_1^T X > \|\theta_1\|_2^2/2\}}$ satisfies*

$$P_{\theta_0}^{(n)} \phi_n \vee P_\theta^{(n)}(1 - \phi_n) \leq e^{-\|\theta_0 - \theta_1\|_2^2/(32\sigma^2)}$$

*for all $\theta \in \mathbb{R}^n$ with $\|\theta - \theta_1\|_2 \leq \|\theta_0 - \theta_1\|_2/2$.*

**Lemma C.2** ((Ghosal & Van Der Vaart, 2007, Lemma 9)). *Suppose $d_n$ and $e_n$ satisfy the above hypothesis testing condition (40). Suppose there exists some nonincreasing function $\epsilon \mapsto N(\epsilon)$ and some $\epsilon_n > 0$ such that for any $\epsilon > \epsilon_n$*

$$\mathcal{N}(\frac{\epsilon\xi}{2}, \{\theta \in \Theta : d_n(\theta, \theta_0) < \epsilon\}, e_n) \leq N(\epsilon).$$

*Then for any $\epsilon > \epsilon_n$, there exists a test $\phi_n$ possibly depending on $\epsilon$ such that*

$$P_{\theta_0}^{(n)} \phi_n \leq N(\epsilon) \frac{e^{-Kn\epsilon^2}}{1 - e^{-Kn\epsilon^2}}, \quad P_\theta^{(n)}(1 - \phi_n) \leq e^{-Kn\epsilon^2 j^2},$$

*for any $j \in \mathbb{N}$ and $\theta \in \Theta$ with $d_n(\theta, \theta_0) > j\epsilon$.*

**Lemma C.3** ((Ghosal & Van Der Vaart, 2007, Lemma 10)). *For every $\epsilon > 0$, denote as*

$$B_n(\theta_0, \epsilon) = \{\theta \in \Theta : \frac{1}{n} \sum_{i=1}^n K(p_{\theta_0,i}, p_{\theta,i}) \leq \epsilon^2, \frac{1}{n} \sum_{i=1}^n V_{2,0}(p_{\theta_0,i}, p_{\theta,i}) \leq \epsilon^2\},$$

*where $K$ and $V_{2,0}$ represent the KL divergence and variance respectively. Then for every probability measure $\bar{\Pi}$ supported on the set $B_n(\theta_0, \epsilon)$, for every $C > 0$,*

$$P_{\theta_0}^{(n)} \left( \int \frac{p_\theta^{(n)}}{p_{\theta_0}^{(n)}} d\bar{\Pi}(\theta) \leq e^{-(1+C)n\epsilon^2} \right) \leq C^{-1}(n\epsilon^2)^{-1}.$$

*Note that in the case that $p_{\theta,i}$ is the density of $\mathcal{N}(y_{\theta,i}, \sigma^2)$, we have*

$$B_n(\theta_0, \epsilon) = \{\theta \in \Theta : \frac{1}{n} \sum_{i=1}^n \frac{|y_{\theta,i} - y_{\theta_0,i}|^2}{\sigma^2} \leq \epsilon^2\}.$$

## C.2. Auxiliary tools for Proposition 4.2

**Lemma C.4** ((Lu et al., 2021a, Theorem 4.5)). *For $x \geq 0$,*

$$x = -\frac{1}{2}[\sigma_3(x + 3) - 5\sigma_3(x + 2) + 7\sigma_3(x + 1) - 3\sigma_3(x) + 6],$$

$$x^2 = -\frac{1}{6}[\sigma_3(x + 2) - 4\sigma_3(x + 1) + 3\sigma_3(x) - 4].$$

**Lemma C.5** ((Lu et al., 2021a, Theorem 4.5)). *For $x, y \geq 0$,*

$$xy = -\frac{1}{12} \big[\sigma_3(x + y - 2) - 4\sigma_3(x + y - 1) + 3\sigma_3(x + y)$$
$$-\sigma_3(x + 2) + 4\sigma_3(x + 1) - 3\sigma_3(x) - \sigma_3(y + 2) + 4\sigma_3(y + 1) - 3\sigma_3(y) + 4\big].$$

## C.3. Auxiliary tools for verifying Lemma 4.1

**Lemma C.6** ((Lu et al., 2021a)). *Let $\theta_1, \theta_2 \in \Theta(L, W, S, B)$. Then we have*

$$\sup_{x \in \Omega} |f_{\theta_1}(x) - f_{\theta_2}(x)| \vee \sup_{x \in \Omega} \|\nabla f_{\theta_1}(x) - \nabla f_{\theta_2}(x)\|_\infty \vee \sup_{x \in \Omega} \|\nabla^2 f_{\theta_1}(x) - \nabla^2 f_{\theta_2}(x)\|_\infty$$
$$= O(W^{\frac{3^{L-1}-1}{2}} (B \vee d)^{\frac{5 \cdot 3^{L-1}-1}{2}}) \cdot \|\theta_1 - \theta_2\|_\infty.$$

## C.4. Auxiliary tools for Proposition 4.3

**Rademacher complexity**   The Rademacher complexity of a given function class $\mathcal{G}$ is given by

$$R_n(\mathcal{G}) := \mathbb{E}\left[\sup_{g \in \mathcal{G}} \left| \frac{1}{n} \sum_{j=1}^{n} \tau_j g(Z_j) \right| \right],$$

where $Z_j$ are i.i.d. samples according to the data distributions and $\tau_j$ are i.i.d. Rademacher random variables.

**Lemma C.7** ((Giné & Nickl, 2021, Theorem 3.3.9)). *Let $\mathcal{F}$ be a countable set of real-valued functions on a measurable space $(S, \mathcal{S}, \mu)$ such that for all $f \in \mathcal{F}$, we have $\|f\|_\infty \leq U < \infty$, $\mathbb{E}_\mu[f] = 0$, and $\mathbb{E}_\mu[f^2] = \sigma^2 \leq U^2$. Let $n \in \mathbb{N}$ and take $X_1, \ldots, X_n \overset{i.i.d.}{\sim} \mu$. Then for any $t > 0$, we have*

$$\sup_{f \in \mathcal{F}} \frac{1}{n} \sum_{i=1}^{n} f(X_i) \leq 2\mathbb{E}\left[\sup_{f \in \mathcal{F}} \frac{1}{n} \sum_{i=1}^{n} f(X_i)\right] + \frac{4Ut}{3n} + \sqrt{\frac{2\sigma^2 t}{n}}$$

*with probability at least $1 - e^{-t}$.*

*Proof.* By (Giné & Nickl, 2021, Theorem 3.3.9), the following holds with probability at least $1 - e^{-t}$

$$\sup_{f \in \mathcal{F}} \frac{1}{n} \sum_{i=1}^{n} f(X_i) < \mathbb{E}\left[\sup_{f \in \mathcal{F}} \frac{1}{n} \sum_{i=1}^{n} f(X_i)\right] + \frac{\sqrt{2v_n t}}{n} + \frac{Ut}{3n},$$

where

$$v_n = 2U\mathbb{E}\left[\sup_{f \in \mathcal{F}} \sum_{i=1}^{n} f(X_i)\right] + n\sigma^2.$$

Then using $\sqrt{a+b} \leq \sqrt{a} + \sqrt{b}$ and AM-GM inequality, we have

$$\sqrt{2v_n t} \leq \sqrt{4U\mathbb{E}\left[\sup_{f \in \mathcal{F}} \sum_{i=1}^{n} f(X_i)\right] t} + \sqrt{2n\sigma^2 t} \leq \mathbb{E}\left[\sup_{f \in \mathcal{F}} \sum_{i=1}^{n} f(X_i)\right] + Ut + \sqrt{2n\sigma^2 t}.$$

Plugging this back and rearranging terms yield the result. $\qquad\square$

**Lemma C.8** ((Maurer, 2016, Corollary 4)). *Let $\mathcal{X}$ be any set, and $(x_1, \ldots, x_n) \subset \mathcal{X}^n$. Let $\mathcal{F}$ be a class of functions $f : \mathcal{X} \to \ell_2$ and let $h_i : \ell_2 \to \mathbb{R}$ be $C_L$-Lipschitz. Then*

$$\mathbb{E}\sup_{f \in \mathcal{F}} \sum_i \epsilon_i h_i(f(x_i)) \leq \sqrt{2} C_L \mathbb{E} \sup_{f \in \mathcal{F}} \sum_{i,k} \epsilon_{ik} f_k(x_i),$$

*where $\epsilon_{ik}$ is an independent doubly indexed Rademacher sequence and $f_k(x_i)$ is the $k$-th component of $f(x_i)$.*

**Lemma C.9** ((Vershynin, 2018, Lemma 6.3.2)). *Let $\mathcal{F}$ be a class of functions and $\{X_i\}_{i=1}^{n}$ be independent and identically distributed random variables. Then*

$$\mathbb{E}\sup_{f \in \mathcal{F}} \left| \frac{1}{n} \sum_{i=1}^{n} f(X_i) - \mathbb{E}[f] \right| \leq 2R_n(\mathcal{F}).$$

**Lemma C.10** ((Lu et al., 2021a, Lemma A.7)). *Consider a function class $\mathcal{F}$ and assume that there exists a sub-root function such that for any $r > 0$*

$$R_n(\{u \in \mathcal{F} : \mathbb{E}[u] \leq r\}) \leq \phi(r).$$

*Then we have*

$$\mathbb{E}\left[\sup_{u \in \mathcal{F}} \sum_{i=1}^{n} \sigma_i \frac{u(X_i)}{\mathbb{E}[u] + r}\right] \leq \frac{4\phi(r)}{r},$$

*where $\{\sigma_i\}_{i=1}^{n}$ are independent Rademacher random variables and $\{X_i\}_{i=1}^{n}$ are independent and identically distributed random variables.*

## D. Construction of the cutoff function

In this section, we construct a $C^2$-cutoff function using $\sigma_3$. Consider a set of nodes $\{t_i\}_{i=1}^8$ satisfying $t_1 = -2, t_2 = -\frac{7}{4}, t_3 = -\frac{3}{2}, t_1 = -1$, and $t_{9-i} = -t_i, i \in [4]$. Define the cutoff function as

$$\phi(x) := b + \sum_{i=1}^8 a_i \sigma_3(x - t_i),$$

for some weights $\{a_i\}_{i=1}^8$ and bias $b$ to be determined. Moreover, we would like $\phi$ to satisfy $\phi(x) = x$, when $-1 < x < 1$, $\phi(x) = -2$ when $x \le -2$, $\phi(x) = 2$ when $x \ge 2$.

Note that when $-1 < x < 1$, $\phi(x) = b + \sum_{i=1}^4 a_i \sigma_3(x - t_i)$. By imposing $\phi(x) = x$ when $-1 < x < 1$, we have

$$\phi'''(x) = 0 \quad \Rightarrow \quad \sum_{i=1}^4 a_i = 0,$$

$$\phi''(x) = 0 \quad \Rightarrow \quad 6 \sum_{i=1}^4 a_i(x - t_i) = 0,$$

$$\phi'(x) = 1 \quad \Rightarrow \quad 3 \sum_{i=1}^4 (x - t_i)^2 = 1.$$

Simplifying these three equations gives

$$\sum_{i=1}^4 a_i = 0, \quad \sum_{i=1}^4 a_i t_i = 0, \quad \sum_{i=1}^4 a_i t_i^2 = \frac{1}{3}.$$

Also we need

$$\phi(-1) = b + \sum_{i=1}^3 a_i(x - t_i)^3 = -1$$

$$\phi(x) = b = -2, \quad x \le -2.$$

Solving the equations above yields

$$b = -2, \quad a_1 = \frac{14}{3}, \quad a_2 = -\frac{32}{3}, \quad , a_3 = \frac{20}{3}, \quad a_4 = -\frac{2}{3}.$$

We further set $a_{9-i} = -a_i, i \in [4]$.

So far, we construct a $C^2$-function $\phi$ such that $\phi(x) = -2$ when $x \le -2$, and $\phi(x) = x$ when $-1 < x < 1$. We now check that $\phi(x) = 2$ on the region $x \ge 2$. By direct calculation, when $x \ge 2$, we have

$$\phi(x) = b + \sum_{i=1}^4 a_i(x - t_i)^3 + \sum_{i=1}^4 (-a_i)(x + t_i)^3$$

$$= b + \sum_{i=1}^4 a_i(-2t_i)(3x^2 + t_i^2)$$

$$= b - 2 \sum_{i=1}^4 a_i t_i^3 = -2,$$

where the third equation follows from the construction that $\sum_{i=1}^4 a_i t_i = 0$.

Given $F > 0$, we can construct a $F$-clipped function by

$$\text{clip}_F(x) := F \cdot \phi(\frac{x}{F}) = bF + \sum_{i=1}^8 a_i F \cdot \sigma_3(\frac{x}{F} - t_i),$$

satisfying $\text{clip}_F(x) = x$ when $-F < x < F$, $\text{clip}_F(x) = 2F$ when $x \ge 2F$, and $\text{clip}_F(x) = -2F$ when $x \le -2F$.

