# OpenReview forum: "Posterior Concentration of Bayesian Physics-Informed Neural Networks for Elliptic PDEs"
_ICML.cc/2026/Conference — ICML 2026 regular_

### Official Review · Reviewer_SEds · 2026-03-03

**Soundness:** 4
**Presentation:** 4
**Significance:** 4
**Originality:** 3
**Overall Recommendation:** 5
**Confidence:** 4

**Summary:**

This paper introduces a Bayesian version of physics-informed neural networks (PINNs). It constructs a Bayesian neural network equipped with a carefully designed prior distribution on the network parameters, and establishes its consistency for solving second-order elliptic PDEs with non-homonegeous Dirichlet boundary conditions. Notably, as the main result of this work, the convergence rate of the population PINN loss of the posterior mean estimator actually matches the minimax optimal rate.

**Compliance With Llm Reviewing Policy:**

Affirmed.

**Final Justification:**

The authors' response has addressed most of my concern.

**Key Questions For Authors:**

1. The cut-off function clip is introduced in order to ensure that the model space $\mathcal{F}(L,W,S,B)$ is uniformly bounded. However, it seems somewhat artificial and lacks a clear theoretical or practical motivation. Is there any related work that adopts a similar setting? What if this setting is discarded?

For instance, by the definition of the prior distribution of the network parameters, is there any chance to prove that the networks without the cut-off function have uniform bound $O(\log n)$ with high probability $1-o(n)$? Does such estimation leads to the same convergence rate as that in Theorem 3.1?

2. How is the network ``trained’’ in this paper? In other words, what method does this work suggest to obtain the posterior mean $\bar{u} _ n$ in Corollary 3.2? I notice that Yang et al. (2021) suggests several methods to sample from the posterior, such as Hamilton Monte Carlo and Variational Inference. Does this paper use the same methods?

3. Since the network setting of this paper (notably, the existence of the cut-off function) is different from related works (Yang et al. 2021 & Sun et al. 2024), adding experimental results would help demonstrate the practical feasibility of the proposed method in this paper.

**Limitations:**

Yes.

**Strengths And Weaknesses:**

Strengths:
1. The minimax optimality achieved by the proposed Bayesian PINN method represents a significant and noteworthy theoretical result.
2. The elliptic PDE considered this paper is formulated in a general form with non-homogeneous boundary condition. Although such setting introduces additional challenges for the estimation of generalization error with respect to stronger norms---as discussed in Remark 3.3, the established convergence rate of the PINN loss and the $H^{1/2}$-generalization error still presents a substantial theoretical contribution.

Weaknesses (discussed later in Key Questions):
1. The cut-function clip, introduced in line 156, appears unnatural.
2. The paper does not clarify the computational procedure for obtaining the proposed Bayesian estimator in practice.
3. There is a notable absence of numerical experiments or simulation studies to validate the theoretical findings.

---

> ### Author Rebuttal · Authors · 2026-03-30
>
> We thank the reviewer for their feedback. We address their questions below.
>
> **Clipping** We thank the reviewer for bringing up this question. We would like to clarify that the existence of this clipping would ensure the uniform boundedness of the output of the neural network class, which plays a crucial role in controlling statistical errors using local Radamacher complexity. To better understand where uniform boundedness is required, see the assumptions of Lemma A.3 and A.5 in Appendix. Although we acknowledge this construction may lower the practical implications of this paper, similar technical issues have arisen and the clipping has been adopted in the theoretical study of posterior contraction for regression problems; see (Lee et al.). The current proof framework would break if we remove this clipping, because the statistical error becomes difficult to bound. It would be appreciated if the reviewer can provide some intuition why $O(\\log n)$ may be achievable.
>
> **Training and experiments** We do not include experiments since this paper focuses on the theoretical aspects of Bayesian PINNs. Validating the precise contraction rate requires efficiently sampling from high dimensional posterior distribution, which is a challenging question on its own; previous papers on posterior contraction for regression problems do not have experiments either; see (Lee et al.) and (Polson and Ročková). We plan to investigate the high dimensional posterior sampling problem in future work.
>
> **References**
>
> [1] Lee K, Lin L, Park J, Jeong S. Posterior contraction for sparse neural networks in besov spaces with intrinsic dimensionality. arXiv preprint arXiv:2506.19144. 2025 Jun 23.
>
> [2] Polson NG, Ročková V. Posterior concentration for sparse deep learning. Advances in neural information processing systems. 2018;31.

---

> > ### Author Rebuttal · Reviewer_SEds · 2026-04-01
> >
> > The authors' response is adequate.

---

### Official Review · Reviewer_5hJN · 2026-03-11

**Soundness:** 2
**Presentation:** 3
**Significance:** 1
**Originality:** 3
**Overall Recommendation:** 3
**Confidence:** 4

**Summary:**

This paper establishes the posterior concentration rate of Bayesian PINNs for elliptic PDEs with non-homogeneous Dirichlet boundary conditions. The authors rigorously prove the relevant theorems (Theorems 3.1 and 3.5) under the given assumptions, where the rate depends on the smoothness parameter $\beta$, the dimensionality $d$, and the numbers of samples $n$, $n_1$, and $n_2$. The theorems show that, under an appropriately constructed prior distribution on the network, the posterior distribution of the model concentrates around the true solution at a near-minimax rate. Moreover, the proposed prior is rate-adaptive.

**Compliance With Llm Reviewing Policy:**

Affirmed.

**Final Justification:**

Dear AC, considering:
1) there are no experiments
2) B-PINN is not widely adopted
3) for curse of dimensionality, the cited paper [1] has a remark: 'There is a common belief that Machine learning based PDE solvers can break the curse of dimensionality, ......, Combined with these approximation bounds, we can also achieve a bound that breaks the curse of dimensionality using Theorem 4.14 and 4.11. In this paper, we aim to consider the statistical power of the loss function in common function spaces and leave the curse of dimensionality as a separate topic.', but the author does not have a similar claim in their rebuttal.

Overall, I will keep my score.

**Key Questions For Authors:**

1. Could you show that the performance of the activation function $\sigma_3$ is equivalent to that of the original B-PINNs, which use the tanh activation function, either theoretically or experimentally?

2. Could you give a definition of “sufficiently smooth”? I disagree with the claim that “$A$ and $V$ are sufficiently smooth, i.e., $A, V \in C^{\infty}(\bar{\Omega})$.”

3. In line 239, should the denominator be “$\int_{\sum_\mathcal{F}}$”? Or what is the integration domain of the denominator?

4. What is the function of $\beta^*$ in Theorem 3.1?

5. Since your given rate has the dimensionality $d$, does it mean the Bayesian Physics-Informed Neural Networks have curse of dimensionality?

Suggestions:

1. To avoid misunderstanding, I suggest using 'Bayesian PINNs' or 'B-PINNs' (as used in the original paper), rather than PINNs, as the abbreviation for Bayesian Physics-Informed Neural Networks in the abstract.

2. The title should be 'Posterior Concentration of Bayesian Physics-Informed Neural Networks for Elliptic PDEs', since this theorem cannot be extended to Physics-Informed Neural Networks.

3. The notation should be unified, for example, $\sum_F$ and $\sum_\mathcal{F}$.

4. Add the proof stop symbol at the end of each proof.

**Limitations:**

yes

**Strengths And Weaknesses:**

Soundness: This paper is a purely theoretical work, and the claims are rigorously supported by theorems, whose proofs appear to be sound. However, there are no experiments to support the theoretical results, although the assumptions seem relatively easy to implement in practice.

Presentation: This paper has good presentation, as the authors summarize the notations and introduce the preliminaries in detail. They also clearly state their objectives at the beginning of the 'Main Results' section and revisit these objectives in the conclusion.

Significance: I have some doubts about the significance, since Bayesian PINNs are not as widely adopted as standard PINNs or the Deep Ritz Method. In addition, the authors use the activation function $\sigma_3$, which is not a commonly used activation function.

Strengths:

1) This paper provides a detailed and concise introduction to the preliminaries.

2) The proofs of both theorems and lemmas are complete and rigorous.

Weaknesses:

1) The activation function $\sigma_3$ used in the paper is not common, which makes the practical contribution less clear.

2) The Bayesian PINN framework used in this work is not widely adopted, which may limit the impact of the results.

---

> ### Author Rebuttal · Authors · 2026-03-30
>
> We thank the reviewer for their feedback. We address their question below.
>
> **ReLU3 activation function** While (Yang et al.) use tanh as their activation function, we believe that ReLU3 is also a suitable choice in the PDE learning literature, particularly when one needs to evaluate the performance of neural network solutions under high-order Sobolev norms, due to its differentiability and ability to efficiently represent high-order polynomials; see (Lu et al.) and (Jiao et al.). On the other hand, given the existence of  approximation theory of neural networks with tanh activation (De Ryck et al.), it is an open but interesting direction to construct a neural network function class and a prior accordingly to
> study the posterior contraction of Bayesian PINNs since this is closer to the setting of (Yang et al.). To validate the exact contraction rate in practice, one need efficient sampling algorithms to explore the high dimensional posterior distribution, which is challenging question on its own. Since posterior contraction has been little understood in the community of Bayesian PINNs, we see our paper as a first step to bridge this gap and leave investigation of those open questions in future work.
>
> **Ambiguity of 'sufficiently smooth'** We apologize for the confusion caused due to the term 'sufficiently smooth'. The assumption we want to make here is that $A, V \\in C^\\infty(\\bar{\\Omega})$, so that the regularity of the PDE solution $u^\\ast \\in C^\\beta(\\bar{\\Omega})$ is induced by the source  $f \\in C^{\\beta-2}(\\bar{\\Omega})$
> and the boundary term $g \\in C^\\beta(\\partial \\Omega)$. See also Remark 3.4 where we discuss the relaxation of this assumption. We will make the assumption on $A, V$ more precise in the revised version.
>
> **Domain of the integral in line 239** We apologize for the confusion, and we would like to clarify that the domain of integral in the dominator should be the entire DNN-function class $\\mathcal{F}$. $\\Sigma_{\\mathcal{F}}$ is the sigma-algebra, on which the prior is defined, and $\\mathcal{F}$ is the largest set inside $\\Sigma_{\\mathcal{F}}$. We will revise this equation to make it clearer.
>
> **Function of $\\beta^\\ast$** We would like to clarify that $\\beta^\\ast$ is the upper bound of all admissible smoothness levels, and it is a crucial hyperparameter to achieve adaptivity, since the construction of the prior depends only on $\\beta^\\ast$, but not on the smoothness $\\beta$ of the PDE solution $u^\\ast$.
> In Theorem 3.1, we show that for any PDE solution $u^\\ast \\in C^\\beta(\\bar{\\Omega})$ satisfying $2 < \\beta \\leq \\beta^\\ast$ and $\\|u^\\ast\\|_{C^\\beta(\\bar{\\Omega})} \\leq K$,  the posterior mass concentrates around the true solution $u^\\ast$ at the (almost) minimax rate $\epsilon_n$. Moreover, with an appropriate chosen prior, our procedure achieves (nearly) optimal posterior contraction for the exact solution over a broad range of smoothness levels, while remaining smoothness-agnostic —— the construction of the prior does not require any knowledge of the underlying smoothness $\\beta$.
>
> **Curse of dimensionality** Yes. We show that posterior mass concentrates around the ground truth at a non-parametric rate $\\epsilon_n$, which suffers from curse of dimensionality. This is consistent with existing Bayesian posterior contraction results for regression problems.
>
> **Writing** We thank the reviewer for their useful suggestions for the presentation. To avoid misunderstanding, we will use 'Bayesian Physics-Informed Neural Network' in the title, and 'Bayesian PINN' as its abbreviation in the abstract. Besides, we will fix the typos in the paper, and add a proof environment in the Section 4 to improve readability.
>
> **References**
>
> [1] Lu Y, Chen H, Lu J, Ying L, Blanchet J. Machine learning for elliptic PDEs: Fast rate generalization bound, neural scaling law and minimax optimality. arXiv preprint arXiv:2110.06897. 2021 Oct 13.
>
> [2] Jiao Y, Lai Y, Li D, Lu X, Wang F, Wang Y, Yang JZ. A rate of convergence of physics informed neural networks for the linear second order elliptic pdes. arXiv preprint arXiv:2109.01780. 2021 Sep 4.
>
> [3] De Ryck T, Lanthaler S, Mishra S. On the approximation of functions by tanh neural networks. Neural Networks. 2021 Nov 1;143:732-50.

---

> > ### Author Rebuttal · Reviewer_5hJN · 2026-04-01
> >
> > I still have two further questions about it.
> >
> > 1) Ambiguity of 'sufficiently smooth'. Do you think that “sufficiently smooth”, such as $A, V \in C^\beta(\bar{\Omega})$, would be enough for your requirements. I think it will make your theorem applicable to a wider range of problems.
> >
> > 2) Curse of dimensionality. Although it is consistent with existing Bayesian posterior contraction results for regression problems, it is not consistent with the well-known advantage of PINNs: tackling the curse of dimensionality. Does this imply that Bayesian-PINNs reintroduce the curse of dimensionality? If not, why is an upper bound that exhibits the curse of dimensionality still valuable for a model that have already tackled it?

---

> > > ### Author Response · Authors · 2026-04-05
> > >
> > > We thank the reviewer for their follow-up questions. We address them below.
> > >
> > > **Ambiguity of 'sufficiently smooth'** Our original intention is to learn PDE solutions corresponding to a fixed elliptic operator (coefficients $A$ and $V$ are fixed and known precisely) given noisy observations of $f$ and $g$, which come from a function class with a broad range of smoothness levels $2 < \\beta \\leq \\beta^\\ast$. Under this setting, it is natural to assume that the coefficients are smooth enough, that is  $A \\in C^{\\beta^\\ast - 1}(\\bar{\\Omega})$ and $V \in C^{\\beta^\\ast - 2}(\\bar{\\Omega})$, such that the regularity of the source term $f \\in C^{\\beta - 2}(\\bar{\\Omega})$ and the boundary term $g \\in C^\\beta(\\bar{\\Omega})$ determines the regularity of the solution $u^\\ast \\in C^\\beta(\\bar{\\Omega})$.
> > >
> > > Back to the reviewer's question, the assumption that $A \in C^{\beta - 1}(\bar{\Omega})$ and $V \in C^{\beta -2}(\bar{\Omega})$ would be enough for the current proof framework. Under this assumption, we can generalize our setup to learning PDE solutions corresponding to a class of second-order elliptic operators. We thank the reviewer for the insight, and we plan to include the relaxed assumption in the revised version.
> > >
> > > **Curse of dimensionality** We thank the reviewer for this interesting question. However,  whether PINNs can overcome the curse of dimensionality depends on the regularity of the PDE solution. If the solution is only in a H\"older space $C^s$ or a Sobolev space $H^s$ with a fixed index $s>0$ which does not grow in the dimension $d$, then there is indeed a curse of dimensionality; see (Lu et al) and (Jiao et al). In fact, the paper by (Lu et al)  also established a lower bound for PINNs with the rate $O(n^{-\\frac{s-2}{d+2s-4}})$ w.r.t $H^2$-norm when the solution lies in $H^s$ for some $s\\geq 2$, which indicates that the curse of dimensionality is unavoidable. On the other hand, if the solution has better regularity, e.g. if the solution is Barron assumed in (Lu, Lu, Wang 2021), then the generalization error of deep Ritz method does not incur a curse of dimensionality; their analysis can be adapted to PINNs under the same Barron assumption. We will comment on this in the revised version.
> > >
> > > Also, we do not agree with that "Bayesian-PINNs introduces  reintroduce the curse of dimensionality". In fact, the rate we obtained for Bayesian PINNs is the same as the rate for PINNs as obtained by (Lu et al). We show that one big advantage of Bayesian-PINNs is that it can achieve rate-adaptivity.
> > >
> > > **References**
> > >
> > > [1] Lu Y, Chen H, Lu J, Ying L, Blanchet J. Machine learning for elliptic PDEs: Fast rate generalization bound, neural scaling law and minimax optimality. arXiv preprint arXiv:2110.06897. 2021 Oct 13.
> > >
> > > [2] Lu Y, Lu J, Wang M. A priori generalization analysis of the deep Ritz method for solving high dimensional elliptic partial differential equations. InConference on learning theory 2021 Jul 21 (pp. 3196-3241). PMLR.
> > >
> > > [3] Jiao Y, Lai Y, Li D, Lu X, Wang F, Wang Y, Yang JZ. A rate of convergence of physics informed neural networks for the linear second order elliptic pdes. arXiv preprint arXiv:2109.01780. 2021 Sep 4.

---

### Official Review · Reviewer_m1uS · 2026-03-13

**Soundness:** 4
**Presentation:** 4
**Significance:** 3
**Originality:** 2
**Overall Recommendation:** 4
**Confidence:** 3

**Summary:**

Physics-Informed Neural Networks (PINNs) approximate the solution to PDEs by training on neural network on a loss that balances matching the boundary conditions with satisfying the PDE. Bayesian PINNs perform Bayesian inference over the parameters of the neural network (i.e., the are “PI-BNNs”), which is useful when there is limited or noisy data. This paper proves that for elliptic PDEs with Dirichlet boundary conditions (e.g., Laplace’s or Poisson’s equation), the posterior of Bayesian PINNs contracts around the ground truth solution a near-minimax optimal rate, without prior knowledge of the smoothness of the ground truth solution.

Assumptions include that the PDE has a solution in a Holder space (so it’s smooth), that a specific spike-and-slab prior is used, that a feedforward architecture is used, …

The prior assumes a width that behaves like $1/(\text{width}!)$ and parameters that have an independent spike-and-slab distribution with the slab component being a uniform distribution of an exponentially distributed length. In other words, the parameters are the width, weights, and bound of the weights prior (so the last is a hyperprior). The depth is fixed to a constant. Unlike the well-cited Polson and Rockova (2018) work, the sparsity is induced by the Bernoulli prior (from the spike-and-slab) over the weights, rather than being explicitly exponentially distributed. Also, the exponential hyperprior over the size of the weights allows them to be unbounded, which the authors note is important to ensure sufficient approximation accuracy of the function-space sieve.

The theorem follows the established approach of showing contraction of the empirical loss and then bounding the generalization error. While the latter can be accomplished by appealing to standard Rademacher complexity results, the former takes more care while, to my knowledge, still following established procedures. The idea is construct a sieve of neural networks that isn’t too big (covering number bounded, shown by bounding it terms of the parameter-space covering number) while also receiving most of the prior mass. The other condition is that the prior places enough mass near the ground truth (specifically, that all functions with empirical loss under a shrinking amount have a lower bounded prior probability, which can be proven by applying a triangle inequality relative to the ground truth).

**Compliance With Llm Reviewing Policy:**

Affirmed.

**Final Justification:**

I appreciate the authors responses regarding novelty and unboundedness assumption on the weights. Overall I think there is value in precise posterior concentration results like this, particularly in this setting of unknown smoothness. Otherwise the assumptions seem fairly reasonable and the I don’t think experimental verification is necessary for this kind of paper. I’ll maintain my score.

**Key Questions For Authors:**

- Have you considered solving the “inverse” problem where the parameters of the PDE are also inferred (A and V, in Equation 1)? Can you hypothesize whether this would change the rates?
- Can you elaborate on why you studied the setting of observing only the source term and the nonhomogeneous boundary data common (and not the solution directly, as in Sun et al.)? Can works studying this setting be cited?
- Can you provide intuition on why the unbounded weights are needed? I understand this would make approximation easier, but I’m (naively, probably) surprised this is needed given the solution is assumed to be smooth. I understand this is not the focus of this paper, but I like to have an eye on whether theoretical analysis of priors would change practical implementations, so things like grading clipping might not allow the weights to get to big.

**Limitations:**

Yes

**Strengths And Weaknesses:**

The paper is well written with a clear explanation of the assumptions and intuition behind the proofs. It is a rigorous theoretical paper on Bayesian consistency of a relatively new Bayesian problem class. Assuming the the ground truth belongs to Holder space is reasonable for the problem setting of elliptical PDE solutions. Obtaining rates that do not depend a prior knowledge of the smoothness is important, as misspecification of smoothness is well-known to induce slow learning rates.

It would be nice to see more citations of Bayesian PINNs being used in practice (the original Yang 2020 has greater than 1k citations so there are plenty).

Regarding the significance, to me the results do not appear too surprising, given the established BNN contraction rates for spike-and-slab priors.

Regarding the originality, the proof techniques seem mostly standard based on my knowledge, but if the authors would like to highlight any especially novel approaches in the rebuttal that would be appreciated.

There are no experiments verifying the results, but I believe this is typical for theory papers of this sort (e.g, the Polson and Rockova (2018) paper does not have experiments either).

---

> ### Author Rebuttal · Authors · 2026-03-30
>
> We thank the reviewer for their feedback. We address their questions below.
>
> **Citations** We thank the reviewer for their kind reminder of the vigorous growth of the Bayesian PINNs community. We will survey the literature carefully and cite more papers in the revised version.
>
> **Novelty of the proof** We would like to highlight here the novelty of our proof. Our spike-and-slab prior is inspired by (Polson and Ročková) but differs from theirs in several important aspects. First, we model sparsity by assigning each neuron an activation probability, which in turn induces a distribution over the overall sparsity level; by contrast, (Polson and Ročková) posits an exponentially distributed sparsity level directly. Second, we place an exponential prior on the amplitude  $B$ of weights, thereby allowing unbounded weights,  while (Polson and Ročková) restricts $B\\leq 1$. The unboundedness of $B$ enables us to take advantage of the approximation capacity of our DNN-class for the target solution (see Proposition 4.2), which will be essential to establish the key ingredients for the posterior contraction rate (see Lemma 4.1). Besides the aforementioned facets, another distinction in the setting of PINNs is that the likelihood term, induced by the empirical PINN-loss,  involves both noisy source and boundary data from different distributions, whereas the likelihood in regression problems admits a empirical square-loss with i.i.d. data. Therefore, we carefully adapt Theorem 4 of (Ghosal and Van Der Vaart) to the setting of Bayesian PINNs (see Lemma 4.1).
>
> **Inverse problems** We thank the reviewer for asking this interesting question. (Sun et al.) studied the forward PDE learning and inverse problem together, and showed that the posterior contracts at the true solution at an (near) optimal nonparametric rate, while they need to know the source function precisely and the parameter in the inverse setting is a constant. Similarly but from an optimization perspective, (Nathan et al.) introduced a kernel estimator with PDE regularization which may enjoy a parametric rate, under the assumption that the PDE operator (including coefficients) and the source term are known precisely. This improved rate has also been suggested by (Sun et al.), although in both papers their estimators are computationally intractable. In our case, while our theoretical analysis can not be easily adapted in the case where only noisy observations of the coefficients $a, V$ are available, we conjecture that the rate of learning the PDE solution may be dominated by the rate of estimating the coefficients. Whether an estimator for a general class of elliptic operators exists such that it outperforms nonparametric rate while being implementable in practice remains an open question. In fact, this has been the focus in one of our ongoing project and we will report results along this line in the future.
>
> **Observing only the source and boundary data instead of the solution directly** We admit that our setting is different from that of Sun et al. However, since we are interested in solving PDEs given limited and noisy information, we believe that only observing the source and boundary is a standard and practical setup, especially in the community of statistical learning of PDEs and physics-informed machine learning; see (Lu et. al) and (De Ryck and Mishra).
>
> **Unbounded weights** As the reviewer mentioned, we need the unbounded weights to comply with the current approximation result (Proposition 4.2). We here discuss one possibility to bound the weights by a uniform constant. A closer look into the proof of Proposition 4.2 shows that the unboundedness is due to the coefficients $\\frac{l^{k-1}}{(k-1)!} \\binom{k}{j}$ in Equation (23) and $\\lambda_{i}(f)$ in Equation (24), and those coefficients make the weights scale as $O(l^d)$. Inspired by Schmidt-Hieber's paper, we can show that $x \mapsto l^d x$ can be implemented by a 2-layer $\\sigma_3$-neural network with $O(l^d)$-width with weights bounded by some uniform constant. However, the price is that the overall width and sparsity of the resulting neural network would scale as $O(l^{2d})$ instead of $O(l^d)$. To accommodate this new approximation result, we may need to construct a new prior to satisfy Lemma 4.1. Since posterior contraction has been little understood in the community of PINNs, we see this paper as a first step to bridge the gap and leave optimization of the result in the future.
>
> **References**
>
> [1] Lu Y, Chen H, Lu J, Ying L, Blanchet J. Machine learning for elliptic PDEs: Fast rate generalization bound, neural scaling law and minimax optimality. arXiv preprint arXiv:2110.06897. 2021 Oct 13.
>
> [2] De Ryck T, Mishra S. Numerical analysis of physics-informed neural networks and related models in physics-informed machine learning. Acta Numerica. 2024 Jul;33:633-713.
>
> [3] Doumèche N, Bach F, Biau G, Boyer C. Physics-informed kernel learning. arXiv preprint arXiv:2409.13786. 2024 Sep 20.

---

> > ### Author Rebuttal · Reviewer_m1uS · 2026-04-03
> >
> > The authors' response is adequate.

---

### Official Review · Reviewer_eE3m · 2026-03-15

**Soundness:** 3
**Presentation:** 3
**Significance:** 2
**Originality:** 3
**Overall Recommendation:** 5
**Confidence:** 3

**Summary:**

This paper proves the (near-minimax) posterior contraction rate of Bayesian Physics-Informed Neural Networks (PINNs) for solving elliptic PDEs with non-homogenous Dirichlet boundary conditions. Based on Bayesian PINN, it adopts spike-and-slab prior for network parameters along with other priors for network width, sparsity, magnitude and structure and establishes a rate-adaptive contraction rate without prior knowledge of the smoothness level of true PDE solution. The proof follows the theoretical framework developed by (Ghosal &
Van Der Vaart, 2017, Chapter 8) and posterior contraction for deep learning (Polson & Rockova, 2018, Section 5). Similar works, e.g. Lu et al. (2021a) and Sun et al. (2024), are properly referenced and compared with difference clearly stated.

**Compliance With Llm Reviewing Policy:**

Affirmed.

**Key Questions For Authors:**

1. In Theorem 3.1, it is assumed that $u^*\in C^{\beta}(\bar\Omega)$. Does "rate-adaptive to smoothness of $u^*$" mean that the posterior contraction rate $\epsilon_n$ depends on $\beta$, but not on $\beta^*$?

2. Line 256: typo -- the first $\infty$ should be 0.

**Limitations:**

The authors adequately discussed the limitations and potential mitigation. They are well understandable.

**Strengths And Weaknesses:**

# Strength

The posterior contraction rate established in this paper is for non-homogeneous Dirichlet boundary problems and is adaptive to the (unknown) smoothness of the exact solution.

# Weakness

As admitted by the authors, the contraction rate is established in weaker $H^s$-norm for $s\leq 1/2$ due to the definition of PINN loss. The rate is near-minimax for the same number of interior and boundary observations.

---

> ### Author Rebuttal · Authors · 2026-03-30
>
> We thank the reviewer for their feedback. We address their questions below.
>
> **Does 'rate-adaptive to smoothness of u' mean that the posterior contraction rate $\epsilon_n$ depends on $\beta$, but not on $\beta^*$?** Yes. We would like to clarify that $\\beta^\\ast$ is the upper bound of all admissible smoothness levels. In Theorem 3.1, we show that for any PDE solution $u^\\ast \in C^\\beta(\\bar{\\Omega})$ satisfying $2 < \\beta \\leq \\beta^\ast$ and $\\|u^\\ast \\|_{C^\\beta(\\bar{\\Omega})} \\leq K$,  the posterior mass concentrates around the true solution $u^\\ast$ at the (almost) minimax rate $\\epsilon_n$, which depends on $\\beta$, but not on $\\beta^\\ast$. Moreover, with an appropriate chosen prior, our procedure achieves (nearly) optimal posterior contraction for the exact solution over a broad range of smoothness levels, while remaining smoothness-agnostic —— the construction of the prior does not require any knowledge of the underlying smoothness $\\beta$.
>
> **Line 256: typo** We thank the reviewer for pointing out that the first infinity should be zero. We will fix this typo in the revised version.

---

> > ### Author Rebuttal · Reviewer_eE3m · 2026-04-06
> >
> > Thanks for the clarification. I do not have other concerns.

---

### Decision · Program_Chairs · 2026-04-30

**Decision:**

Accept (regular)

**Comment:**

This paper proves an interesting result for Bayesian PINNs for a general class of elliptic problems, namely the posterior concentrates around the exact solution at a near optimal minimax rate and this rate *adapts* with respect to the smoothness of the solution. It assumes sufficient smoothness of the solution and a suitable prior. However, these assumptions are not overtly restrictive. The reviewers have acknowledged the theoretical correctness of the paper in terms of the proofs as well as novelties in the proof. In terms of weaknesses, the paper is a purely theoretical contribution with no experimental evidence and Bayesian PINNs are unlikely to be widely used in practice, given all the problems with PINNs that are increasingly coming to the fore. Nevertheless, given the solid theoretical contribution of the paper in an area relevant to the scientific machine learning community, the paper is an interesting addition to the literature.